

**Unraveling biogeographical patterns and environmental drivers of soil**
**fungal diversity at the French national scale**
Christophe Djemiel[1], Samuel Dequiedt[1], Walid Horrigue[1], Arthur Bailly[1], Mélanie Lelièvre[1],
Julie Tripied[1], Charles Guilland[1,2], Solène Perrin[1], Gwendoline Comment[1], Nicolas P.A. Saby[3],
Claudy Jolivet[3], Antonio Bispo[3], Line Boulonne[3], Antoine Pierart[4], Patrick Wincker[5], Corinne
Cruaud[5], Pierre-Alain Maron[1], Sébastien Terrat[1] and Lionel Ranjard[1]
[1] Agroécologie, INRAE, Institut Agro, Univ. Bourgogne Franche-Comté, F-21000 Dijon,
France
[2] Novasol experts, Dijon, France (present address)
[3] INRAE, Info&Sols, F-45075 Orléans, France
[4] ADEME, Service Agriculture et Forêt, Angers, France
[5] Génomique Métabolique, Genoscope, Institut François Jacob, CEA, CNRS, Univ Evry,
Université Paris-Saclay, 91057, Evry, France
**ORCID**
Christophe DJEMIEL [0000-0002-5659-7876]
Walid HORRIGUE [0000-0001-5834-9983]
Nicolas SABY 0000-0001-8964-1194
Claudy JOLIVET 0000-0001-5871-6413
Antonio BISPO 0000-0002-6985-8385
Line BOULONNE [0000-0002-5789-6691]
Patrick WINCKER 0000-0001-7562-3454
Corinne CRUAUD 0000-0002-4752-7278
Pierre-Alain MARON [0000-0003-2315-0741]
Sébastien TERRAT [0000-0001-5209-6196]
Lionel RANJARD [0000-0002-7720-5843]
*Correspondance to:* Lionel Ranjard (lionel.ranjard@inrae.fr)



**Abstract**
The fungal kingdom is among the most diversified kingdoms on earth with estimations up to
12 million species. Yet, it remains poorly understood with only 150,000 fungal species currently
described. Given the major ecological role of fungi in ecosystem functioning, these numbers
stress the importance of investigating fungal diversity description across different ecosystem
types. Here, we explored the spatial distribution of the soil fungal diversity on a broad
geographical scale, using the French Soil Quality Monitoring Network that covers the whole
French territory (2,171 soils sampled along a systematic grid). Fungal alpha-diversity was
assessed directly from soil DNA using a metabarcoding approach. Total cumulated fungal
diversity across France included 136,219 OTUs, i.e., about 1% of the global soil fungal
diversity for a territory representing only 0.3% of terrestrial surface on Earth. Based on this
dataset, the first extensive map of fungal alpha-diversity was drawn and evidenced a
heterogeneous and spatially structured distribution in large biogeographical patterns of 231 km
radius for richness (Hill number q=0) and smaller patterns of 36 km radius for dominant fungi
(Hill number q=2). As related to other environmental parameters, the spatial distribution of
fungal diversity was mainly influenced by local filters such as soil characteristics and land
management, but also by global filters such as climate conditions. The spatial distribution of
abundant and rare fungi was determined by distinct or similar filters with various relative
influences. Interestingly, cropped soils exhibited the highest pool of fungal diversity relatively
to forest and vineyard soils. In complement, soil fungal OTUs network interactions were
calculated under the different land uses across France. They varied hugely and showed a loss
of 75% of the complexity in crop systems and grasslands compared to forests, and up to 83%
in vineyard systems. Overall, our study revealed that a nation-wide survey with a high spatial
resolution approach is relevant to deeply investigate the spatial distribution and determinism of
soil fungal diversity. Our findings provide novel insights for a better understanding of soil
fungal ecology and upgrade biodiversity conservation policies by supplying representative
repositories dedicated to soil microorganisms in a context of global change.



## 1. **Introduction**

The fungal kingdom has been evolving continuously for more than 800 million years to adapt to and colonize a large number of habitats on Earth (Loron et al., 2019; Naranjo-Ortiz and Gabaldón, 2019; Li et al., 2021; Bonneville et al., 2020; Berbee et al., 2020). This heterotrophic kingdom represents about 2% of the global biomass on Earth (Bar-On et al., 2018) and is among the most diverse kingdom in the Eukaryota domain (Mora et al., 2011; Blackwell, 2011; Taylor et al., 2014; Hawksworth and Lücking, 2017). Recent extrapolations based on environmental DNA characterization using a metabarcoding approaches evaluated that the total number of fungal taxa ranged from 6.28 million to 12 million (Baldrian et al., 2022; Wu et al., 2019; Phukhamsakda et al., 2022). To date, only 150,000 fungal species have been described by the scientific community to date (Species Fungorum 2022, http://www.speciesfungorum.org/Names/Names.asp), namely only 1.25-2.4% of the whole estimated fungal diversity.

The majority of fungi is found in terrestrial ecosystems, especially in soils; only 4,000 extant fungi from freshwater habitats are presently listed (Calabon et al., 2022). Fungi play crucial ecological roles in soils by contributing to their proper functioning due to a wide range of functional guilds (Bar-On et al., 2018; Frac et al., 2018). Fungi are keystones of the soil food web, both in biogeochemical cycles and in interactions with other macro- and microorganisms (de Vries et al., 2013; Treseder and Lennon, 2015; Hannula and Träger, 2020). The functions of fungal communities provide many ecosystem services that promote mineral nutrition of plants linked to soil organic matter rates and nitrogen availability (Miyauchi et al., 2020; Ward et al., 2022). They are important decomposers (saprotrophs) of organic matter: they break down complex biopolymers and play a key role in organic matter recycling (Hage and Rosso, 2021). Regarding symbiotic interactions, there are no less than 50,000 mycorrhizal fungi interacting with 340,000 land plants representing 90% of beneficial symbiosis with a host plant association (Genre et al., 2020). Fungal pathogens and parasites can cause diseases and important crop losses and have a significant economic impact (Möller and Stukenbrock, 2017; Fausto et al., 2019). Some of them are also identified as biocontrol agents and involved in plant protection through the regulation of pathogenic microorganisms (Peng et al., 2021). Fungi are also known as ecosystem engineers contributing to soil aggregation and maintenance. Thus, the stability of the soil structure largely depends on mycelium density and the pool of fungal enzymatic activities (Lehmann et al., 2020).

The spatial distribution of fungal diversity has been more recently and less studied than the spatial distribution of bacterial diversity, in particular when it comes to identifying and



ranking the local and global filters that influence species richness (Griffiths et al., 2011; Terrat
et al., 2017; Fierer and Jackson, 2006; Delgado-Baquerizo et al., 2018; Ranjard et al., 2013). In
2014, Tedersoo and collaborators generated the first global map of fungal richness and revealed
that fungi are spatially structured and exhibit biogeographical patterns (Tedersoo et al., 2014).
One major hotspot of fungal diversity was located in Latin America and a cold-spot in the Sahel
region, whereas fungal diversity distribution was shown homogeneous across Europe. A second
global map of fungal diversity was drawn from a meta-analysis of the *chao* index (Větrovský
et al., 2019a). Conversely to Tedersoo and collaborators (Tedersoo et al., 2014), they
highlighted a lower fungal diversity in Latin America and a higher in North Africa (Větrovský
et al., 2019a). In parallel, Ma and collaborators (Ma et al., 2017) revealed a heterogeneous
spatial distribution according to a gradient of forest soils at the continental scale in Eastern
China. These differences and even discrepancies in the location of biogeographical patterns at
global scale reflect the huge gap of knowledge in fungal diversity distribution and the need to
complete these studies with high resolution approaches at a finer scale.
Soil fungal communities are constantly subjected to natural biotic and abiotic stresses,
but also to human activities through global warming, deforestation, and land use intensification.
These stresses altogether have a significant influence on fungal abundance, diversity and
community assembly and lead to an overall impact on soil functions (Pärtel et al., 2017; Geisen
et al., 2019; Thomson et al., 2015; Tsiafouli et al., 2015; Shi et al., 2019). Several large- and
small-scale studies showed that fungal richness is driven by land uses, edaphic factors, climate
conditions, and spatial descriptors (Tedersoo et al., 2014; Ma et al., 2017; Větrovský et al.,
2019a; George et al., 2019a). Farming practices on cropped soils such as tillage, fertilization or
crop rotation can have an influence on fungal diversity (Sommermann et al., 2018; Sadet-
Bourgeteau et al., 2019; Stefan et al., 2021; Finn et al., 2021) or not (Lentendu et al., 2014).
Altogether, soil pH emerges as the strongest driver of fungal alpha-diversity, similar to bacteria
at global or territorial scales (Delgado-Baquerizo et al., 2018; Terrat et al., 2017). Some studies
indicate a positive effect of soil pH on fungal richness at the national (George et al., 2019b) and
global (Bastida et al., 2021) scales, whereas others show a negative effect on a global scale
(Tedersoo et al., 2014). In 2020, Tedersoo and collaborators also proposed a unimodal
relationship between soil pH and fungal richness / Shannon diversity (Tedersoo et al., 2020).
Fungal richness also appears higher in fine soil textures than in coarse soil textures (George et
al., 2019b). In the same vein, plant cover – especially increased tree richness – shows a positive
impact on fungal richness and Shannon diversity (Tedersoo et al., 2020). Soil carbon is also an
important driver of fungal diversity, with a positive (Maestre et al., 2015; Yang et al., 2019;



Bastida et al., 2021) or a negative effect (George et al., 2019b), depending on studies. Other
findings show that soil calcium and phosphorus, the C:N ratio, bulk density, some spatial
descriptors (e.g., latitude, longitude, altitude) and climate conditions also have an influence on
fungal alpha-diversity, yet to a lesser extent (Tedersoo et al., 2014; Maestre et al., 2015; Bastida
et al., 2021).
In the face of these discrepancies and given the essential role of fungi for ecosystem functioning
and sustainability, it is essential to deeply characterize soil fungal diversity – in terms of alpha-
and beta-diversity –, using the most recent molecular and high-throughput methods to better
decipher the impacts of global and local filters (Chu et al., 2020; Hyde, 2022). In this context,
we investigated the French Soil Quality Monitoring Network (RMQS) using a metabarcoding
approach to determine the soil fungal diversity at a national scale. Based on a regular grid of
2,171 sites across France, this survey captured the various land uses, climates, geomorphology
types and soil characteristics. Located in Western Europe, France is the third largest European
country and exhibits the third highest pedological diversity across the world according to the
WRB classification (Minasny et al., 2010) and is also known to exhibit diversified land use and
climate conditions (Karimi et al., 2020a; Ballabio et al., 2016). Main land uses in France are
dominated by croplands, grasslands and forests. Climate conditions are also among the most
diverse ones in Europe and organized in three major poles: oceanic, Mediterranean, and
mountainous. All these statements lead us to conclude that France could be considered as an
ideal national-scale observatory for monitoring the variations of biotic and abiotic components
of soil ecosystems. The RMQS soil sampling strategy is probably one of the most intensive and
extensive national soil sampling strategy in the world and the systematic random sampling leads
to good spatial coverage profitable both for mapping soil characteristics and unraveling
environmental variation. Using this soil survey, a substantial body of scientific knowledge on
soil bacterial biogeography has been produced by the use of molecular tools (Ranjard et al.,
2013; Terrat et al., 2017; Karimi et al., 2018, 2020b), associated with several technical
developments to standardize metabarcoding, associated bioinformatics and statistical analysis
(Djemiel et al., 2020; Terrat et al., 2019, 2012).

156        Here, we used high-throughput sequencing to study the small subunit 18S rDNA gene

directly amplified from soil DNA to characterize the soil fungal diversity. In most studies,
alpha-diversity is characterized by one index. However, in order to obtain a global overview of
fungal diversity in a biogeographical context, it is important to explore deeper the distribution
of rare and dominant taxa and ranking the influence of environmental filters in a deterministic
process (Jousset et al., 2017; Rivett and Bell, 2018; Jiao and Lu, 2020a). To reach this objective



we used Hill numbers to combine complementary diversity indexes such as richness, Shannon
diversity and inverse Simpson. Methods based on spatial prediction, (geostatistics) were applied
to the data to map and analyze the macro-ecological patterns of soil fungal diversity along the
environmental gradients encountered in France. We used a set of environmental datasets –
including soil physico-chemical characteristics, climate conditions and land use – to explain
variations in soil fungal diversity and rank the ecological processes and the environmental
filters structuring the spatial fungal distribution on a wide extent. *In fine*, we compared the
variations of fungal diversity across different land uses with the variations of the complexity of
fungal interactions network by inferring co-occurrence networks at the operational taxonomic
unit (OTU) level.

2.   **Methods**
2.1. Soil sampling design
Soils were sampled from 2,171 locations across France between June 2000 and June 2009 as
part of the RMQS set up to monitor the quality of French soils. As described previously
(Ranjard et al., 2013), these sites included a wide range of land uses: forests (n=589), grasslands
(n=537), crops (n=886), vineyards/orchards (n=65) and low anthropized environments (n=94),
and eight climatic regions. Inside each of the 2,171 cells of a regular 16 km × 16 km grid
throughout France, a smaller 20 m × 20 m grid was used for sampling where 25 core samples
of topsoil (approximately 0-30 cm depth) were taken. The core samples were bulked to obtain
a composite sample. The sampling protocols applied on the RMQS are available through the
RMQS2 manual (Jolivet et al., 2022). Each sample was air-dried and sieved to 2 mm and
separated into two sub-samples. The first sub-sample was frozen at -40 ºC for molecular
analyses, while the second sub-sample was used for physico-chemical analyses. A detailed
description of the physico-chemical analyses performed in this study is accessible from (Jolivet
et   al.,   2006).   All   the   data   are   available   in   dataverse   Gis   Sol
(https://doi.org/10.15454/QSXKGA).

2.2. Molecular characterization of fungal communities

191         2.2.1.  Soil DNA extraction

Soil DNA was extracted from 1g of soil using the GnS-GII standard procedure (Terrat et al.,
2012, 2015). Briefly, the soil underwent two lysis steps, i.e., mechanical lysis and chemical
lysis. In both cases, the soil was ground and homogenized for 90 s with 2 g of 0.1 mm diameter
silica beads, 2.5 g of 1.4 mm diameter ceramic beads and 4 glass beads of 4 mm diameter in 5



ml of a mix solution containing 100 mM Tris-HCl (pH 8), 100 mM EDTA (pH 8), 100 mM
NaCl, 2% (wt/vol) sodium dodecyl sulfate and up to 2.5 ml ultrapure water using a Fast-Prep-
24 classic kit, and then incubated at 70 °C for 30 min. The mixture was centrifuged (7,000 g,
20 °C, 5 min) to retrieve the lysate. A deproteinization step was necessary using 1/10[th] of
volume with 3 M potassium acetate (pH 5.5) followed by a centrifugation step (14,000 g, 4 °C,
5 min) to recover the supernatant containing the soil DNA. DNA was precipitated using
isopropanol at -20 °C, and stored at -20 °C for 30 min. The last step consisted in washing the
DNA pellet with ethanol and resuspending it in 200 μl of ultrapure water. Then, crude DNAs
were purified with NucleoSpin Soil kits following the manufacturer's instructions (NucleoSpin
Soil, Macherey-Nagel). The purified DNAs were quantified by fluorescence (QuantiFluor,
Promega) using an Infinite® 200 PRO plate reader (Tecan) and then normalized to 5 ng.

2.2.2.  Library preparation for sequencing
The V7-V8 regions of the fungal 18S rDNA gene were amplified from purified DNAs using
forward primer FR1 and reverse primer FF390 (Chemidlin Prévost-Bouré et al., 2011) with a
two-step PCR. Both amplifications were carried out in a total volume of 25 μl using 1 to 5 ng
of DNA, 4 μl of 5x HOT FIREPol® Blend Master Mix with 7.5 mM $MgCl_2$ (Solis Biodyne,
Tartu, Estonia), and 1 μl (10 μM) of each primer (Eurogentec). The first step amplified the
target region under the following conditions: initial denaturation at 94 °C for 3 min, followed
by 35 cycles of 94 °C for 30 s; 52 °C for 1 m; 72 °C for 1 mn, and final extension at 72 °C for
5 min. The 18S PCR products were purified using an AMPure bead kit (Beckman) and
quantified using a QuantiFluor staining kit (Promega, USA). The second amplification was
performed to add barcodes for multiplexing samples. The conditions of the second PCR were
similar, with a reduced number of cycles (seven) and a specific purification with a MinElute kit
(Qiagen). Library preparation for the 2,171 samples was conducted at the GenoSol platform;
Illumina HiSeq 2 × 250 bp paired-end sequencing was conducted by Genoscope (Evry, France).

2.3. Bioinformatic analysis
We used BIOCOM-PIPE v.20 pipeline (https://forgemia.inra.fr/biocom/biocom-pipe) to
process the 18S rRNA gene sequences (Djemiel et al., 2020). FASTQ paired-end raw reads
were filtered with PRINSEQ to keep the good-quality sequences and then overlapped with
FLASH to form contiguous reads. The libraries were demultiplexed and trimmed with zero
difference between the barcode and primer sequences. The sequences were aligned with the
Infernal tool based on RNA structures (Nawrocki and Eddy, 2013). Chimeras were removed by



a "hunting-recovering" step specific to BIOCOM-PIPE (Djemiel et al., 2020). Following this,
a global clustering at 95% similarity was performed with the cleaned sequences, to cluster into
Operational Taxonomic Units (or OTUs), followed by a post-clustering step with ReClustOR
to improve the clustering (Terrat et al., 2019). All our diversity indices (geostatistical modeling,
variance partitioning) and co-occurrence networks analyzes are performed on this OTU-based
approaches with a post-clustering step for consistence with the evaluations of fungal diversity
generally described in the literature.

238       2.4. Statistical analysis

Hill numbers were calculated to estimate alpha-diversity and compare samples on a linear scale
to provide a complete interpretation of alpha-diversity through different metrics. Hill numbers
with $q = 0$ corresponded to the OTU richness observed in a sample (emphasizes rare fungal
OTUs), $q = 1$ to the exponential of Shannon diversity (correspond to "typical" or "common"
fungal OTUs), and $q = 2$ to inverse Simpson index (correspond to dominant fungal OTUs)
(Alberdi and Gilbert, 2019). We tested if the variables were normally distributed or
approximately so, using the *shapiro.test* function. Depending on the result, we applied a boxcox
transformation if the gaussian assumption was not satisfied. To compute the boxcox
transformation from forecast package (Hyndman et al., 2020), we estimated the lambda value
with the *BoxCox.lambda* function and applied the transformation with the *BoxCox* function.
Outliers were tracked using the *grubbs.test* function in the "outliers" package (Komsta and
Komsta, 2011) for each Hill number dataset. Once the outliers were removed, to estimate the
multiple comparisons across the modalities (land uses or climate types), analyses of variance
(ANOVAs) were used, and we verified the normality assumption of residuals. If it was satisfied,
we used a Least Significant Difference (LSD) test with adjusted p-value < 0.05; if it was not
satisfied, we used a nonparametric test with *kruskal* function from "agricolae" package and a
correction by the Bonferroni method for the multiple comparisons.
The details of the samples removed in the different land uses are available in Supplementary
Figure 1, as recommended by (Dini-Andreote et al., 2021).
The environmental data that did not follow a Gaussian distribution were log-transformed.
Moreover, the soil pH covaries with land uses, with a majority of acidic soils in forests and a
majority of neutral and basic soils in croplands and vineyards/orchards (Supplementary Figure
2). For all these reasons, we transformed the soil pH variable with a polynomial transformation
of degree 2.



Relationship between fungal alpha-diversity and environmental filters was assessed using
variance partitioning. Briefly, the first step consisted in reducing the effect of model collinearity
to obtain the most parsimonious models. We used the *vif* function (the variance inflation factors
(VIF)) in the "usdm" package (Naimi, 2015) and kept the explanatory variables with VIF ≤ 5.
A second filtering step was performed to determine the best environmental variables using the
*regsubset* function ("leaps" package (Lumley and Lumley, 2013)) and based on the Bayesian
information criterion (BIC) and adjusted $R^2$. Lastly, we conducted a redundancy analysis
(Legendre, 2018) to model variation of the overall environmental filters using the *rda* and
*ordiR2step* functions ("vegan" package (Oksanen et al., 2013)). To select the best variables, we
performed a forward selection to build a model maximizing the adjusted $R^2$. We used the
*ordiR2step* function with 10,000 permutations maximum and the *anova.cca* function ("vegan"
package) to evaluate the variance explained by the best explanatory environmental filtering
variables.
Geostatistical modeling was used to assess the alpha-diversity spatial variations. We followed
a standard approach as proposed in (Granger et al., 2015). First, a variogram model was fitted
to the experimental variogram computed using alpha-diversity observed at the sample sites.
Then, we predicted the unsampled positions by kriging method using a local neighborhood. We
implemented this approach with the "gstat" package (Bivand et al., 2015). We tried to fit various
authorized variogram models and kept the one that minimized the objective function. Then, we
used the results of leave-one-out cross-validation (LOOCV) to evaluate the performance of the
best fitted geostatistical model by computing the standardized squared prediction errors (Lark,

284      2002).

To obtain the information of the putative OTU richness across France, we computed the
rarefaction (interpolation) and prediction (extrapolation) curves for fungal richness (q=0) using
the R package iNEXT (Hsieh et al., 2016).

2.5. Fungal co-occurrence networks
We used the methodological analysis previously described in (Karimi et al., 2020b, 2019) to
compute the fungal co-occurrence networks between land uses. Briefly, two main steps were
required: i) standardizing the number of soils (fixed at 60 samples) used to compute the network
*per* land use to avoid a sampling size effect, and ii) carry out network repetitions (100
repetitions) to integrate the residual heterogeneity of the soils within each land use. Thus, the
minimum number of combinations ensured that each network was computed from a unique
combination of sites. Then, for each replicate, network computation was based on a contingency



matrix of 136,219 fungal OTUs for the 60 randomly selected soil samples. The Spearman
correlation coefficient for each pair of OTUs was used as a similarity index to estimate fungal
OTU co-occurrence. A correlation was considered robust and non-random if the p-value was
below 0.06 after correction using the False-Discovery Rate method. To describe the topology
of the networks, a set of metrics was calculated using the "statnet" package (Handcock et al.,
2019), including the number of connected nodes, the proportion of connected nodes, the number
of links, and the connectance. These metrics are defined in (Karimi et al., 2017a). We used a
*kruskal* test with a correction by the Bonferroni method for the multiple comparisons of the
fungal networks based on land uses, using 100 repetitions. The networks were mapped using
Cytoscape version 3.9.1.

**3. Results**
3.1. Evaluation and extrapolation of fungal alpha-diversity across France
Based on 18S rDNA amplicon sequencing to characterize the fungal diversity of the 2,171
sampled soils, we obtained a total of 180 million raw sequences. After applying different
bioinformatic filters (using BIOCOM-PIPE workflow), we validated fungal diversity on 2,060
samples. Regarding the cumulated fungal diversity from the whole France, we identified
136,219 OTUs from the 2,060 samples. Thanks to our intensive soil sampling strategy
combined with in-depth sequencing, we extrapolated with iNEXT a total of 186,794 OTUs (Fig.
1) at the national scale.

3.2. Spatial distribution of fungal alpha-diversity across France
We generated three national maps showing the soil fungal alpha-diversity for all Hill numbers
using a kriging interpolation approach (Fig. 2). The results of the LOOCV show very low $R^2$
values equal to 0.058, 0.057 and 0.038 for q=0, q=1 and q=2, respectively. However, the median
and the mean of the SSPES are very close to the expected values (e.g., 0.45 and 1). The fitted
variograms reveal different spatial structuring depending on the weighting of OTU relative
abundances (Supplementary Table 1). Thus, the predicted map of fungal richness (Hill number
q=0) exhibited a heterogeneous and spatially structured distribution with a large autocorrelation
distance of about 231-km radius (Fig. 2a, 2d). More or less wide regions with hot or cold spots
of fungal diversity were observed. More specifically, soils from the north-west to the center of
France support a high fungal richness whereas soils from the north-east, the south-east and the
southwest support a lower fungal richness.




The models fitted for q=1 (exp. Shannon diversity) and q=2 (inverse Simpson) exhibited the
spottiest distribution with short autocorrelation distances (27-km radius and 36-km radius,
respectively) (Fig. 2b, 2e and 2c, 2f). The hotspots observed for q=1 and q=2 were less diffuse
and remained strongly present in the north-west to the center of France. This spotty distribution
highlighted small hotspots of abundant fungal OTUs in certain geographical zones described as
having low fungal diversity by q = 0, such as the south-east and the north-east of France.

337        3.3. Relationship between sets of environmental filters and fungal alpha-diversity
We used a variance partitioning approach to evaluate the relative share of fungal diversity
explained variance by each set of environmental variables (soil characteristics, land use, climate
conditions and spatial descriptors) for the different Hill numbers in partial models using a
redundancy analysis (RDA). Globally, environmental filters explained 20.1%, 15.52, % and
7.54% of the total variance of fungal richness for q=0, q=1 and q=2, respectively (Fig. 3a, 4a,
5a and Supplementary Table 2). For q=0, the main drivers of fungal richness variance were the
soil characteristics (11.30%), and then to a lesser extent climate conditions (0.88%), land use
(0.39%) and spatial descriptors (0.26%) (Fig. 3a). For q=1, the soil characteristics (9.30%) were
the main drivers, and then to a lesser extent, spatial descriptors (0.90%), climate conditions
(0.63%) and land use (0.44%) (Fig. 4a). For q=2, the soil parameters (4.25%) and land use
(2.16%) were the main drivers, followed to a lesser extent by climate conditions (0.69%) and
spatial descriptors (0.31%) (Fig. 5a). The percentage of interactions between the environmental
filters decreased from 7.25% for q=0 to 4,25% for q=1 and neared zero for q=2 (0.12%). The
main soil physical and chemical properties for each land use and climate condition are
summarized in Supplementary Figures 3 and 4.

354        3.4. Influence of the soil characteristics on soil fungal alpha-diversity
The key determining soil parameter for fungal richness was the pH (6.72%). A unimodal
relationship was evidenced, with minimum fungal diversity in the most acidic and alkaline soils
(Supplementary Figure 2). For q=0, a significant influence of the soil texture was also
demonstrated, with a linear negative relationship observed with clay (2.38%) and silt (0.65%).
Conversely, a weak positive relationship was observed with the total lead content (0.58%) and
available phosphorus (0.08%) (Fig. 3b).
For q=1, the soil pH (6.30%) had a comparable unimodal distribution and was also the strongest
driver followed by the clay content (3%) with a negative linear correlation (Fig. 4b). For q=2,





the soil pH remained the strongest driver with also a unimodal relationship, but organic carbon
and the total iron content were also identified, with a negative linear relationship (Fig. 5b).

3.5. Influence of climate conditions on soil fungal alpha-diversity
The great diversity of climate conditions in France allowed us to compare fungal diversity
across eight types of climates (Fig. 6a). Temperature, rainfall, and elevation are summarized in
Supplementary Figure 4. Our analyses revealed that fungal diversity for q=0 and q=1 was
highest under oceanic climate (type 5) and lowest under Mediterranean climate (types 6 and 8)
and under the climate of the southwestern basin (type 7) (Fig. 6b and 6c). Fungal diversities for
q=0 and q=1 under mountain and continental climates had an intermediate value between
oceanic and Mediterranean climates. Interestingly, there was no significant difference in fungal
diversity based on dominant OTUs (q=2) across the various climate types of France (Fig. 6d).

3.6. Variation of soil fungal diversity as related to land uses
By comparing fungal richness (q=0) across the different land uses encountered in France (Fig.
7a), we observed several significant differences (Fig. 7b). Forest and vineyard/orchard soils
harbored lower fungal richness than grassland and agricultural soils and can be ranked as
follows: vineyards/orchards (x = 1,384 OTUs) ≤ forests (x = 1,393 OTUs) < grasslands (x =
1,469 OTUs) ≤ crops (x = 1,498 OTUs) (Fig. 7b). The same trend was observed for q=1 (Fig.
7c) but was different for q=2 (Fig. 7d). For q=2, fungal diversity in grassland and vineyard soils
appeared lowest compared to forest and crop soils. Furthermore, the extreme fungal diversity
values greatly varied according to the land use, whatever the metrics used.
Within the four major land uses of French soils, we identified and compared more precisely
land managements (Fig. 8). For example, forests can be categorized into three groups –
deciduous forests, coniferous forests, and mixed forests. Among forest managements, we
observed significant differences between deciduous and mixed forests: the lowest richness was
found in mixed forests (Fig. 8a). For q=1, fungal diversity in deciduous forests was significantly
higher than in coniferous and mixed forests (Fig. 8b), while no difference was detected for q=2
(Fig. 8c). Significant differences were also recorded by comparing the different land
managements of crop systems: soil fungal diversity was higher under crops with grassland
rotation whatever the metrics. No significant difference was recorded between vineyards and
orchards or between the various grassland managements (Fig. 8).

3.7. Comparison of soil fungal co-occurrence networks between land uses



The networks were graphically composed of connections (links) between the nodes corresponding to the OTUs. The links represented the significant positive and negative correlations between the OTUs occurring in the soils under the respective land uses (Fig. 9). A visual analysis of the networks obtained for the different land uses revealed a significant shift in structure ranging from a highly connected, tightly closed structure for forests to a sparse, open structure for vineyards (Fig. 9). In grassland and crop soils, the networks exhibited an intermediate complexity of the structure in terms of number of links and connected OTUs (Supplementary Figure 5). Statistical comparisons of the network metrics between the land uses confirmed a highly significant decreasing gradient of network complexity, with forest >> grassland ≥ crop system > vineyard and orchard soils (Supplementary table 3, Supplementary Figure 5). The average number of links significantly decreased by 84% from forest to vineyard soils and by 76% from forest to crop and grassland soils. The average connectance also progressively decreased by 81% from forest to vineyard soils and by 78% from forest to crop and grassland soils.



## 4. **Discussion**


The first predictions of worldwide fungal diversity ranked from 2,2 to 3,8 million
species (Hawksworth and Lücking, 2017), but recent molecular works updated estimations up
to 6,28 to 12 million species predicted by computing several hundreds of international studies
(Phukhamsakda et al., 2022; Wu et al., 2019; Baldrian et al., 2022). At the national scale, this
question remains unexplored for soil ecosystems. In our study, we predict total fungal richness
at a national scale for the first time by an extrapolation analysis from metabarcoding at the OTU
levels. Compared to the estimated worldwide diversity, France exhibits a very high cumulated
soil fungal richness (of about 1%) relative to its small surface (0.3 % of terrestrial land). This
suggests that global soil fungal diversity is strongly under-estimated worldwide mainly due to
the poor intensive sampling strategy that has been to date only extensive with few sampling
sites. Consequently, this strategy seems relatively inefficient to capture the local environmental
heterogeneity that hosts and shapes fungal richness. Therefore, it is important to gather several
deeply investigated national surveys to estimate global soil fungal diversity more robustly
(Dini-Andreote et al., 2021). Another example of soil fungal diversity estimation at a national
scale has been described in Wales, where 437 samples were collected sites on a surface area of
$20 \times 10^3$ km$^2$, leading to a total evaluation of 4,408 OTUs (George et al., 2019c). For other soil
organisms, total bacterial richness has been evaluated to reach a total of 188,030 OTUs across
France (Terrat et al., 2019). Earthworm richness has been evaluated more globally by compiling
6,928 sites in 57 countries, leading to an estimated 1,376 cumulated species across the world
(Phillips et al., 2019).
Our first maps of the three Hill numbers were provided to describe the spatial distribution of
soil fungal alpha-diversity across France, as previously done for molecular microbial biomass
and bacterial richness (Dequiedt et al., 2011; Terrat et al., 2017). The heterogeneous spatial
distribution of fungal diversity observed in France is not congruent with several studies
(Tedersoo et al., 2014) (Větrovský et al., 2019b), which did not observe significant variations
across Europe with a global soil mapping approach. The two studies computed 365 and 3,085
soil samples across the world, respectively, compared to 2,171 in the present study at the scale
of France. In addition, these global sampling strategies were based on non-random designs that
generally left aside difficult access (polar, arid, mountainous) regions, hence possible biases on
the environmental representativeness of soil fungal habitats. A recent study in northern Europe
generated an extrapolated fungal richness map of Estonia that confirmed a heterogeneous
geographic distribution with hot- and cold-spots at the national scale (Tedersoo et al., 2020).
Altogether, these observations stress the need to assess more intensive samplings at different



scales in order to describe robustly the global distribution of soil fungal diversity and its
determinism (Dini-Andreote et al., 2021).
The community of soil microorganisms – especially fungi – is well known to be largely
dominated by a few highly abundant taxa and to include a large number of rare taxa (Fuhrman,
2009; Pedrós-Alió, 2012; Egidi et al., 2019; Bickel and Or, 2021). Few biogeography studies
have focused on abundant and rare taxa through the different alpha-diversity metrics (e.g., Hill
numbers) (Bent and Forney, 2008). Yet, comparing the spatial distribution of dominant and rare
biosphere fungi is important to better grasp the environmental determinism that shapes soil
fungal diversity (Mo et al., 2018). Mapping of richness (q=0, including all OTUs), "typical"
(q=1, including common fungal OTUs abundances) and dominant (q=2, including OTUs with
high relative abundance) fungal OTUs revealed different spatial patterns, "patchier" (i.e.,
spatially more diffuse) for q=0 with 231 km radius and "spottier" (i.e. spatially more restricted)
for q=1 (27 km) and q=2 (36 km). A similar observation was made in eastern China, with
different spatial distributions of rare and dominant soil fungal OTUs, suggesting differential
sensitivity to various environmental filters leading to increase the endemicity of particular
dominant taxa (Jiao and Lu, 2020b).
The decrease of the explained variance between the q=0, q=1 and q=2 Hill numbers indicates
that environmental and spatial characteristics had a low influence on the national distribution
of dominant OTUs. These are generally considered as generalist and more driven by stochastic
processes, whereas rare taxa are more driven by deterministic processes (Zhao et al., 2022; Jia
et al., 2018; Xu et al., 2022b, a). The spottier distribution observed for q=2 could support this
hypothesis of a more random distribution across France, less influenced by environmental
filters. Whatever the Hill numbers, the main filters explaining the variance of fungal diversity
were the soil characteristics: the soil pH was the main driver, followed by the clay content for
q=0 and q=1, and the trophic conditions for q=2 (organic C and total Fe contents). Such an
influence of soil trophic resources on dominant fungal OTU diversity seems in accordance with
their generalist and copiotrophic strategy (Wang et al., 2021). Numerous studies have reported
the importance of the pH in the distribution of fungal richness across different scales (Rousk et
al., 2010; Tedersoo et al., 2014; Glassman et al., 2017; Tedersoo et al., 2020; George et al.,
2019d). Interestingly, we revealed a unimodal relationship of the soil pH with fungal diversity,
whereas most studies found either a positive or a negative effect (Tedersoo et al., 2014; Maestre
et al., 2015; Bastida et al., 2021; Yang et al., 2019). Such a discrepancy could be partly
explained by the large pH range recorded in France – 3.7 to 9 –, *versus* 3.6 to 5.2 in Wales
(George et al., 2019d) or more surprisingly 3.34 to 10.43 at the European scale (Fernandez-



Ugalde et al., 2022). Emphasizing our hypothesis, other studies report the same unimodal
relationship for fungi and even for bacteria within the same pH range (Bickel et al., 2019;
Tedersoo et al., 2020). As for bacterial richness across France, fungal richness was lower in
fine-textured soil, which is not congruent with the results obtained in Wales (George et al.,
2019c). In France, we may think that fine-textured soils offer less favorable habitats for fungi,
as previously reported (Witzgall et al., 2021; Tecon and Or, 2017). This might be partly
explained by the decrease in microscale heterogeneity with increasing clay content, leading to
a lower diversity of microbial habitats and a smaller hosting capacity for various indigenous
microbial species (Tecon and Or, 2017). Finally, some soil heavy metals were minor but
represented significant drivers of fungal richness. A positive relationship was observed with
total lead, but a negative one with cadmium and nickel. These metallic elements occur naturally
but also result from human activities and are known to be toxic for soil microorganisms when
accumulated in the environment (Sun et al., 2022; Ding et al., 2022). In crop soils, significant
Cd accumulation through the input of phosphate fertilizers extracted from contaminated
limestone rocks has been observed (Khan et al., 2017) and our results could reflect the
significant impact of this contamination on soil microorganisms at a broad scale.
Conversely to bacterial biogeography at the scale of France, climate conditions have been
identified as important global filters of the distribution of fungal diversity across France
whatever the Hill number (Terrat et al., 2017). The highest fungal richness and highest typical
OTU diversity found under oceanic climate may be partly explained by particular conditions
such as buffered mean temperature and humidity inducing soil homeothermy and stability of
water availability favorable to fungal development (Canini et al., 2019; Jiao et al., 2021). On
the contrary, the high variability of these conditions between seasons could explain the decline
observed under Mediterranean climate. The poor influence of climate on the diversity of
abundant fungal OTUs across France could reflect their generalist strategy better adapted to a
high magnitude of environmental fluctuations over time, as previously observed in eastern
China (Jiao and Lu, 2020b). Moreover, our results indicate that rare fungi were more present in
geographical regions with abundant annual rainfall and mild mean temperature, in line with the
observation of increasing fungal richness with frequent rainfall (Tedersoo et al., 2014; Wang et
al., 2018; Bahram et al., 2018).
In France, each land use corresponds to a particular intensity of soil disturbance resulting from
more or less important human activities. We can rank the different land uses according to the
intensity level of their soil disturbance as follows: forests < grasslands < crops <
vineyards/orchards. We observed the highest richness (q=0) and typical (q=1) OTU diversity



in grasslands and crops corresponding to the intermediate levels of disturbance. Similar
observations have been reported about nematodes (Vazquez et al., 2019), bacteria (Delgado-
Baquerizo et al., 2018; Terrat et al., 2017) and fungi (George et al., 2019c) at different scales.
Conversely, no difference has been reported between forest, grassland and crop soils in Estonia
(Tedersoo et al., 2020), and decreased fungal richness has been reported between temperate-
forest and crop soils at a global scale (Bastida et al., 2021). Our observations support the
intermediate disturbance hypothesis (IDH) stating that the species richness of an ecosystem is
maximized when it is submitted to an intermediate disturbance, and minimized when it is
submitted to either a low disturbance by a competitive exclusion process or to a high
disturbance by a selection process (Connell, 1978; Wilkinson, 1999; Giller et al., 1998). More
precisely, agricultural practices such as tillage can stimulate microbial richness in crop systems
(Szoboszlay et al., 2017; Lienhard et al., 2013), and the highest level of richness is generally
explained by the coexistence of microorganisms with different ecological strategies that
promote ecosystem stability (Griffiths and Philippot, 2013). Interestingly, we also observed the
highest diversity of dominant fungal OTUs (inverse Simpson, Hill q=2) in crop systems, *versus*
lowest diversity in grasslands, in line with previous studies reporting a similar trend in grassland
and crop soils (Xu et al., 2017; Zhang et al., 2022).
Analyzing fungal diversity according to the different land managements within each land use
highlighted the highest fungal diversity in the agricultural systems when crop rotation included
grasslands. Inserting temporary grasslands in the rotations is well known to improve soil quality
in terms of nutrient provision and recycling, soil structure and biological regulation (Martin et
al., 2020) and could favor the development of soil microorganisms, as previously described at
the landscape scale in Brittany (Western France, Le Guillou et al., 2019). More fundamentally,
this statement also raises the question of the influence of aboveground (plant) diversity on the
abundance and diversity of belowground (micro-) organisms due to the maintenance of diverse
habitats in soils and to changes in nutrient cycling poorly investigated in crop systems to date
(Wardle et al., 2004). Among forest ecosystems, deciduous forests seem to provide the most
favorable conditions for fungal diversity. Across France, deciduous forests present the highest
tree family richness compared to mixed and coniferous forests (data not showed). Therefore,
our results are in line with studies showing that plant species richness positively affects the soil
fungal diversity (Tedersoo et al., 2016; Hiiesalu et al., 2017). The lowest fungal diversity
observed in coniferous forest soil also confirmed the strong influence of the lower availability
and/or degradability of organic substrates provided by this litter for microorganisms (Leckie et
al., 2004; De Boer et al., 2005). However, other parameters such as plant genotype, forest stand



age or tree density, not taken into account in the present study, could also affect fungal diversity
(Tedersoo et al., 2016; Hazard and Johnson, 2018; Spake et al., 2015). In contrast, no difference
related to the different types of grassland or to the distinction between vineyards and orchards
was recorded.
Beyond fungal alpha-diversity, the analysis of co-occurrence networks is a relevant way of
providing a more comprehensive view of fungal diversity and its interactions according to
environmental variations on a broad scale (Karimi et al., 2017b). As previously observed for
bacterial co-occurrence networks across France, land use intensity affects the complexity of
fungal networks (Karimi et al., 2019). Although forest ecosystems exhibited the lowest fungal
richness, they harbored the highest complexity of fungal interaction networks. Strong losses of
about 83% of the links between forests and vineyards and about 75% between 'forests' on the
one hand and 'grasslands and crop systems' on the other hand were observed across France. A
similar trend has been observed for bacterial networks across France (Karimi et al., 2019) and
also for fungi along a transect from forest to vineyards in Australia (Xue et al., 2022). However,
soil fungal interaction networks remain poorly described by comparing land uses on a broad
scale. The lowest fungal diversity and lowest complexity of interaction networks observed in
vineyard and orchard soils could be related to the intensification of agricultural practices in
these systems. Vineyard soils are indeed known to be strongly disturbed by intensive tillage, a
restricted plant cover and large pesticide inputs (Quiquerez et al., 2022). This intensification of
agricultural practices can lead to the isolation of fungal taxa and the loss of links between taxa
in these soils by i) reducing microbial biomass, hence a lower probability of each cell
encountering another and interacting with it (Dequiedt et al., 2011), ii) stimulating self-
sufficient opportunistic microorganisms that do not interact with others (Lienhard et al., 2013),
and iii) reducing spatial connectivity between soil ecological niches due to soil tillage and
compaction, hence physical isolation of fungal taxa (Cordero and Datta, 2016). Altogether, our
results confirm that forest soils remain a favorable habitat for soil fungi by representing a
mosaic of connected ecological niches that are fully complete and shared by non-opportunistic
taxa (Karimi et al., 2019).

5.  Conclusions
Our study confirms that a nation-wide survey is relevant to deeply investigate the spatial
distribution and determinism of soil fungal diversity. The multiplication and the sum of such
studies conducted across the world could highly upgrade biodiversity conservation policies and
provide representative repositories dedicated to soil microorganisms in a context of global



change. To go further, it will be important to explore fungal beta-diversity and fungal taxonomy
at the scale of France in order to reach a more comprehensive understanding of spatial
distribution, ecological processes and environmental filters. Finally, it will also be important to
investigate the ecological and functional traits assignment of soil fungal communities, using
recent tools and databases developed to better predict the shift in soil functioning according to
land management intensity (Djemiel et al., 2022).

**Code and data availability**
The fungal DNA sequencing datasets supporting the results presented in this article are
available at the EBI ENA under accession number PRJEB57875. The code that supports the
findings of this study are available from the first author upon request.

**Authors' contributions**
All authors conceptualized the research project. Claudy Jolivet coordinated the RMQS program
and the soil sampling at the territory scale. Lionel Ranjard is the scientific coordinator of the
different projects dealing with the characterization of the soil microbial communities at the
scale of France. Charles Guilland, Solène Perrin, Gwendoline Comment, Julie Tripied and
Mélanie Lelièvre performed the molecular analyses. Patrick Wincker and Corinne Cruaud
contributed to DNA sequencing. Christophe Djemiel and Sébastien Terrat performed the
bioinformatic analyses. Claudy Jolivet, Nicolas P.A. Saby, Line Boulonne provided the
environmental dataset. Christophe Djemiel, Samuel Dequiedt, Walid Horrigue, Arthur Bailly,
and Nicolas P.A. Saby contributed to the statistical and geostatistics analyses. Christophe
Djemiel and Lionel Ranjard wrote the original draft. Christophe Djemiel, Sébastien Terrat,
Nicolas P.A. Saby, Claudy Jolivet, Line Boulonne, Antoine Pierart, Pierre-Alain Maron, and
Lionel Ranjard reviewed and edited the final manuscript.

**Competing interests**
One author is member of the editorial board of SOIL journal. The peer-review process was
guided by an independent editor, and the authors have also no other competing interests to
declare.

**Acknowledgements**
This study was granted by "France Génomique" (project number ANR-10-INBS-09-08) and by



the project AgroEcoSol coordinated by Aurea Agrosciences in partnership with INRAE and
ARVALIS. The project is supported by ADEME as part of the Eco-efficient Industry and
Agriculture program of the Future Investments Program (project number 1782C0109). We also
thank the French Biodiversity Agency, the Roullier endowment fund and TMCE company
which financed part of this valuation. In addition, due to the involvement of technical facilities
of the GenoSol platform (DOI 10.15454/L7QN45) of the infrastructure ANAEE-Services,
received a grant from the French state through the National Agency for Research under the
program "Investments for the Future" (reference ANR-11-INBS-0001), as well as a grant from
the Regional Council of Bourgogne Franche Comté. The BRC GenoSol is a part of BRC4Env
(10.15454/TRBJTB), the pillar "Environmental Resources" of the Research Infrastructure
AgroBRC-RARe. RMQS soil sampling and physico-chemical analyses were supported by a
French Scientific Group of Interest on soils: the "GIS Sol", involving the French Ministry for
an Ecological Transition and Territorial Cohesion (MCT), the French Ministry of Agriculture
and Food (MASA), the French Agency for Ecological Transition (ADEME), the French
Biodiversity Agency (OFB), the National Institute of Geographic and Forest Information
(IGN), the French Institute for Research and Development (IRD), the French Geological Survey
(BRGM) and the National Research Institute for Agriculture, Food and Environment (INRAE).
We thank all the soil surveyors and technical assistants involved in sampling the sites and the
staff of the European Soil Samples Conservatory (INRAE Orléans) for preparing the soil
samples. Calculations were performed using HPC resources from DNUM CCUB (Centre de
Calcul de l'Université de Bourgogne). Thanks, are also extended to Frédérick Gavory for
submitting dataset to EBI ENA, and Annie Buchwalter for correcting and improving English
language in the manuscript.

**Financial support**
This study was granted by "France Génomique" (project number ANR-10-INBS-09-08) and by
the project AgroEcoSol coordinated by Aurea Agrosciences in partnership with INRAE and
ARVALIS. The project is supported by ADEME as part of the Eco-efficient Industry and
Agriculture program of the Future Investments Program (project number 1782C0109). We also
thank the French Biodiversity Agency, the Roullier endowment fund and TMCE company
which financed part of this valuation.

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





**Figures**

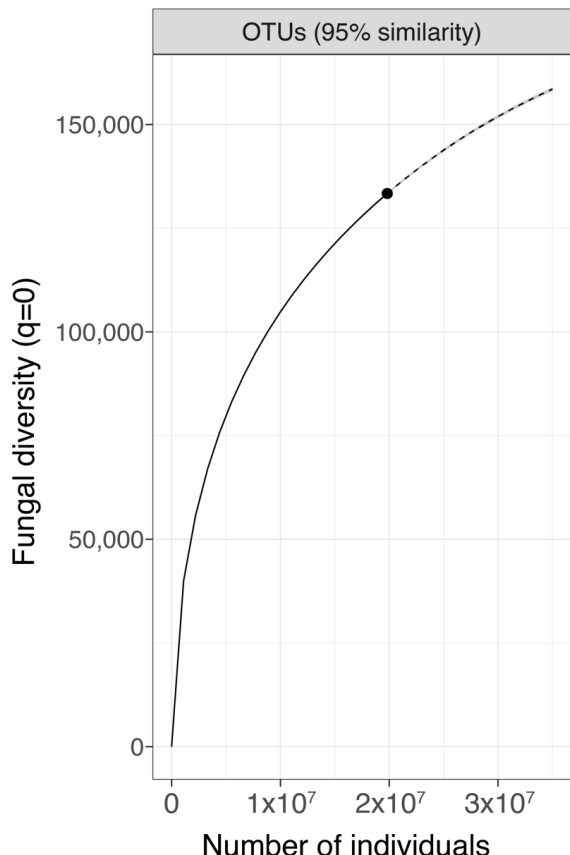

**Figure 1**


Fig. 1: Maximum number of fungal species expected across France based on OTUs. The
analyses of fungal richness rarefaction (solid line segment) and extrapolation (dotted line
segment) was performed with iNEXT.




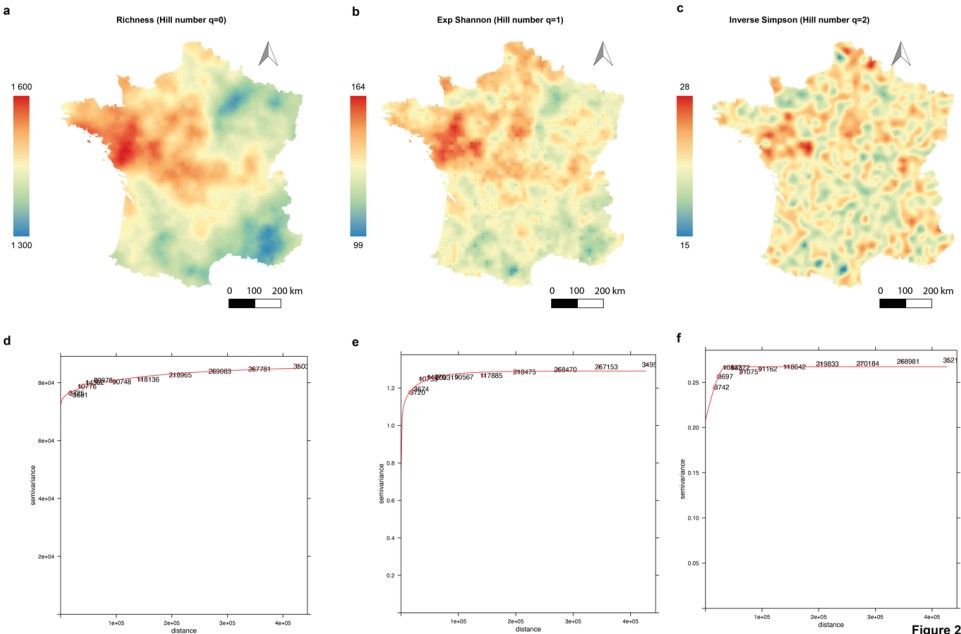


Fig. 2: National soil fungal alpha-diversity maps and robust variograms in France. Map of
fungal richness (a). Map of fungal exponential of Shannon diversity (b). Map of fungal inverse
Simpson (c). The variogram of fungal richness and the exponential of Shannon diversity are
based on Matern model with M. Stein's parameterization (d and e). The variogram of fungal
inverse Simpson is based on a spherical model. Colors correspond to the extrapolated values
expressed as OTUs *per* soil sample (f).

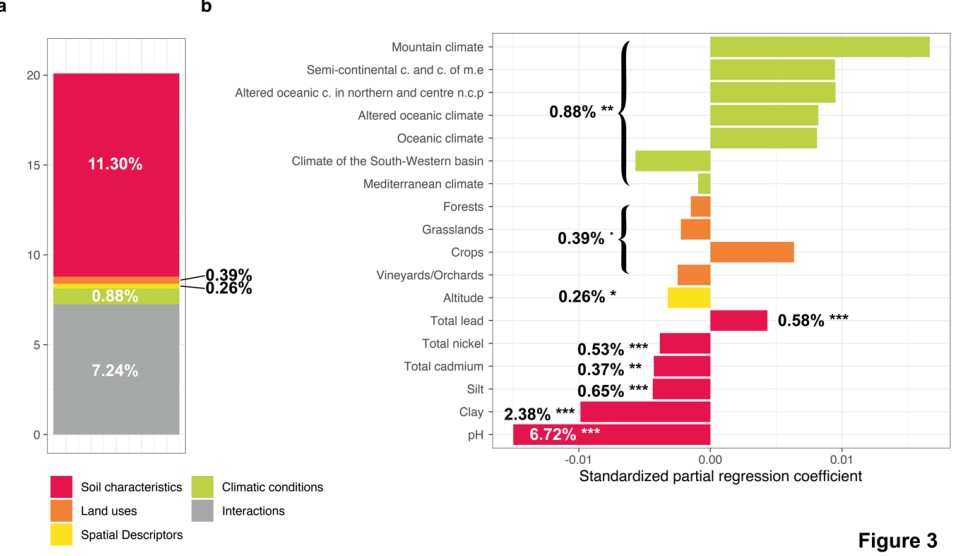




Fig. 3: Variance partitioning analysis to determine how local factors and factors related to
global environmental filters explained variance in fungal richness (a). The amount of explained
variance corresponds to the adjusted $R^2$ values of the contextual groups using partial
redundancy analysis. (b) Model parameters for the distribution of fungal richness. Each
parameter is presented with its estimated model coefficients and its marginal effect assessed by
a permutation test. *P<0.1; **P<0.01; ***P<0.001. Missing values indicate that the variable
was not retained in the model. Sand was removed prior to model evaluation since it was
represented by the opposite of the sum of the silt and clay contents.

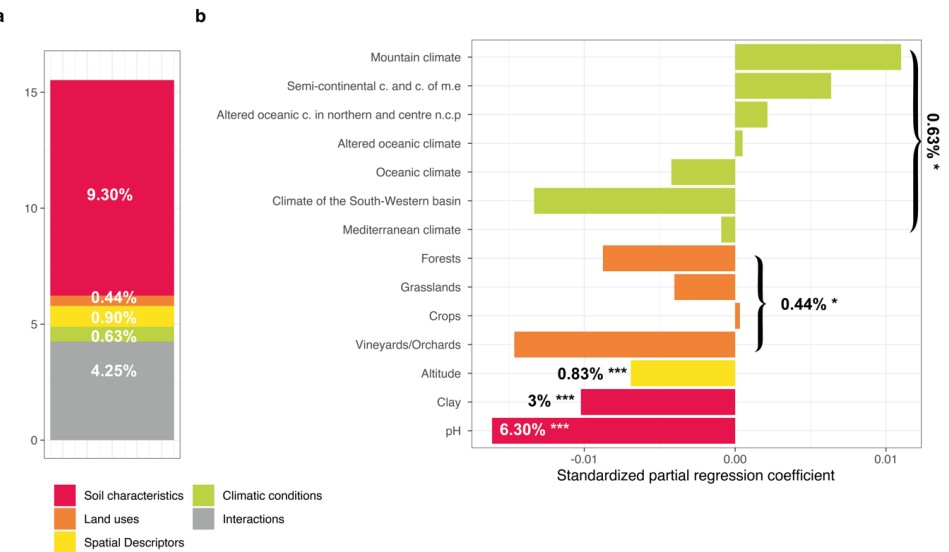

**Figure 4**


Fig. 4: Variance partitioning analysis to determine how local factors and factors related to
global environmental filters explained variance in the exponential of fungal Shannon diversity
(a). The amount of explained variance corresponds to the adjusted $R^2$ values of the contextual
groups using partial redundancy analysis. Model parameters for the distribution of the
exponential of fungal Shannon diversity (b). Each parameter is presented with its estimated
model coefficients and its marginal effect assessed by a permutation test. *P<0.1; **P<0.01;
***P<0.001. Missing values indicate that the variable was not retained in the model. Sand was
removed prior to model evaluation since it was represented by the opposite of the sum of the
silt and clay contents.



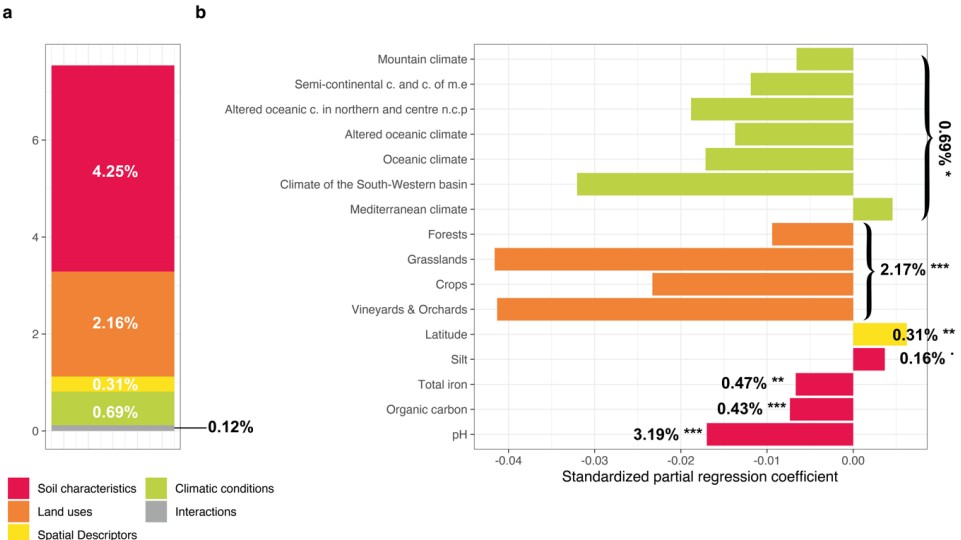

Figure 5


Fig. 5: Variance partitioning analysis to determine how local factors and factors related to
global environmental filters explained variance in fungal inverse Simpson (a). The amount of
explained variance corresponds to the adjusted $R^2$ values of the contextual groups using partial
redundancy analysis. Model parameters for the distribution of fungal inverse Simpson (b). Each
parameter is presented with its estimated model coefficients and its marginal effect assessed by
a permutation test. *P<0.1; **P<0.01; ***P<0.001. Missing values indicate that the variable
was not retained in the model. Sand was removed prior to model evaluation since it was
represented by the opposite of the sum of the silt and clay contents.

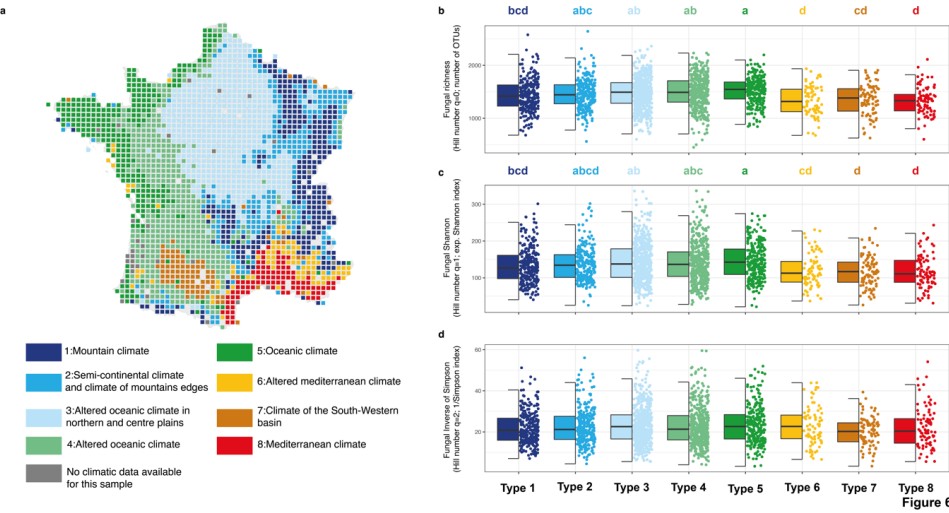




Fig. 6: Map of the RMQS sampling sites and classification of the eight climate types (a). Soil
fungal alpha-diversity distribution across climate types for fungal richness (b), the exponential
of fungal Shannon diversity (c), and fungal inverse Simpson (d). Different letters designate
significantly different values following multiple comparisons.

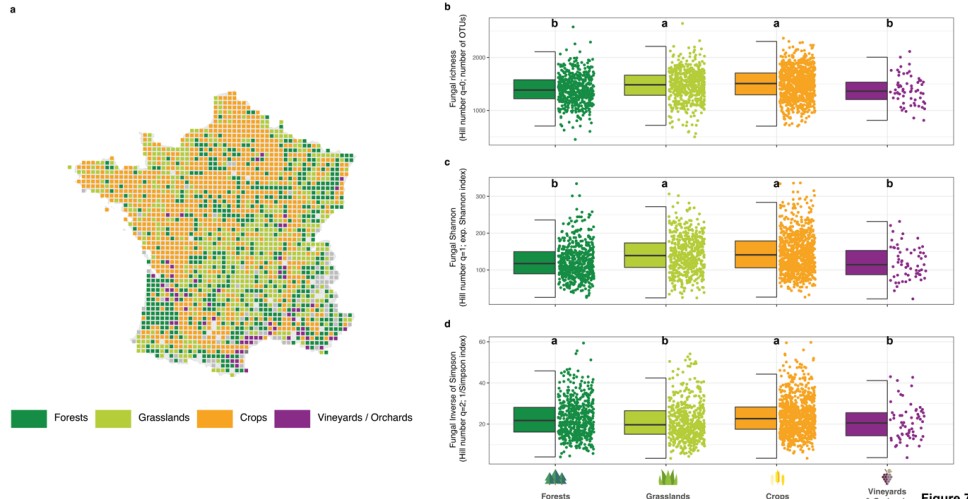


Fig. 7: Map of the RMQS sampling sites and classification for the four land uses (a). Fungal
richness distribution across land uses (b). Distribution of the exponential of fungal Shannon
diversity across land uses (c). Fungal inverse Simpson distribution across land uses (d).
Different letters designate significantly different values following multiple comparisons.

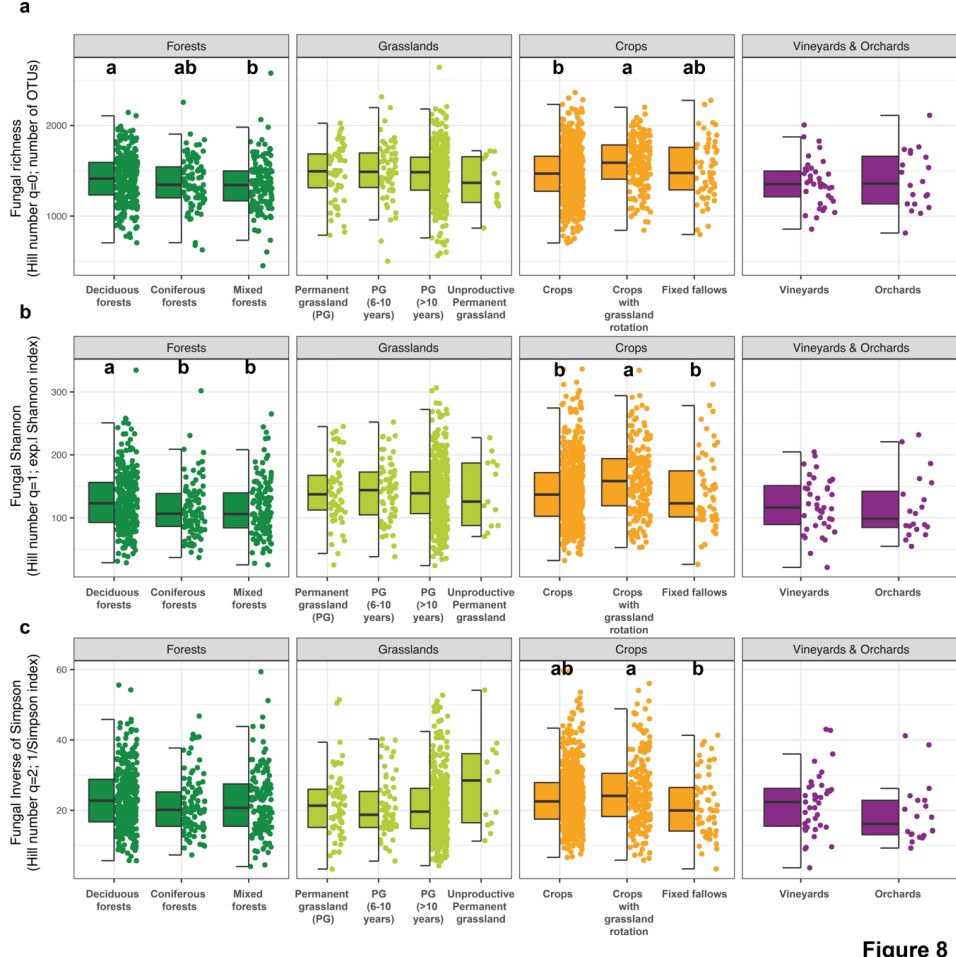

**Figure 8**

Fig. 8: Distribution of soil fungal alpha-diversity within the four major land uses of French soils according to a more precise land management characterization for fungal richness (a), the exponential of fungal Shannon diversity (b), and fungal inverse Simpson (c). Different letters designate significantly different values following multiple comparisons.



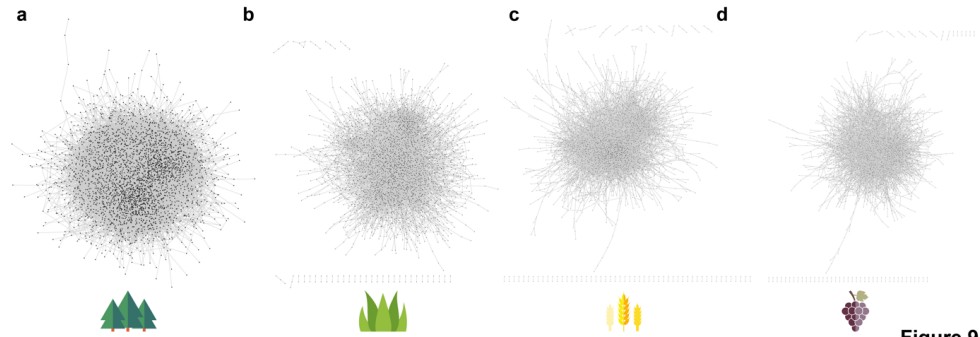

Figure 9


Fig 9. Co-occurrence networks of fungal OTUs across land uses in France. Among the 100
replicates of the 4 types of land cover, we visualized the network closest to the median network
based on the number of links and connectance. Nodes, OTUs; edges, links between the nodes.