# Peer review of "Unraveling biogeographical patterns and environmental drivers of soil fungal diversity at the French national scale 2 3 Christophe Djemiel1, Samuel Dequiedt1, Walid Horrigue1, Arthur Bailly1, Mélanie Lelièvre1, 4 5 Julie Tripied1</sup"

_EGUsphere, 2023_

## Author Response (AR1)

Author's response

Handling topic editor: Elizabeth Bach, elizabeth.bach@tnc.org

We greatly thank the two reviewers for taking time to read our manuscript carefully and providing many insightful comments to improve it.

Reviewer 1
The manuscript entitled "Unraveling biogeographical patterns and environmental drivers of soil fungal diversity at the French national scale", by Djemiel and co-authors investigated soil fungal diversity across a vast sampling in France. The article is well written and analyzed, and makes use of a huge amount of data… sadly, the selected molecular marker is not good/precise enough to elucidate fungal diversity.

We agree with the reviewer that ITS is the most common and historical molecular marker for assessing fungal taxonomy, especially to describe taxonomic groups. However, 18S rDNA has also been an alternative in the last few years to assess the soil fungal community structure and richness more globally, as demonstrated by several authors – in particular in biogeography and at large scales (see below).

- George, P. B., Creer, S., Griffiths, R. I., Emmett, B. A., Robinson, D. A., & Jones, D. L. (2019). Primer and database choice affect fungal functional but not biological diversity findings in a national soil survey. Frontiers in Environmental Science, 173.

- Li, J., Delgado-Baquerizo, M., Wang, J. T., Hu, H. W., Cai, Z. J., Zhu, Y. N., & Singh, B. K. (2019). Fungal richness contributes to multifunctionality in boreal forest soil. Soil Biology and Biochemistry, 136, 107526.

- Bastida, F., Eldridge, D. J., García, C., Kenny Png, G., Bardgett, R. D., & Delgado-Baquerizo, M. (2021). Soil microbial diversity–biomass relationships are driven by soil carbon content across global biomes. The ISME Journal, 15(7), 2081-2091.

- Fernandes, M. L. P., Bastida, F., Jehmlich, N., Martinović, T., Větrovský, T., Baldrian, P., ... & Starke, R. (2022). Functional soil mycobiome across ecosystems. *Journal of Proteomics*, *252*, 104428.

We will add a specific paragraph in the discussion of the manuscript justifying the relevance of our molecular marker, and more broadly about the metabarcoding approach (L432-L433).

***Relevance and drawbacks of the metabarcoding approach***
*Two molecular markers are commonly used to explore fungal diversity thanks to metabarcoding approaches: the internal transcribed spacer (ITS) region, accepted as a universal barcode, and the 18S rRNA gene as an alternative. Both have advantages and drawbacks, in particular to observe specific functional groups. For example, members of the class Glomeromycetes are better characterized using 18S rDNA than ITS, especially in the soil microbiota, and this could have a significant impact on fungal diversity metrics (George et al., 2019). In addition, important reference sequences are only annotated at the phylum level in the international databases (Nilsson et al., 2012; Nilsson et al., 2016; Banos et al., 2018). Moreover, a recent study highlighted that national-scale fungal biogeography studies based on 18S rDNA were relevant to decipher the relationships between fungal diversity and environmental filters (George et al., 2019). For all these reasons, we chose to use the 18S rDNA gene to characterize*

*fungal diversity. Once the molecular marker was chosen, the hypervariable region had to be selected for sequencing. Various criteria had to be taken into account such as amplicon length in relation to the sequencer, or the desired taxonomic and phylogenetic resolution. We sequenced the V7-V8 regions because they appeared to be the most promising regions for fungal diversity assessment (Banos et al., 2018). The last tricky step of a fungal diversity study is the bioinformatics analysis, which is dependent on the pipeline (Pauvert et al., 2019). Nevertheless, our previous studies on bacterial biogeography showed that our metabarcoding pipeline and associated tools were highly appropriate to analyze large microbial datasets (Djemiel et al., 2020; Terrat et al., 2019).*

*Although alpha-diversity depends on the intensity and strategy of sampling, especially when addressing the complexity of the soil microbiota (Willis et al., 2019; Castle et al., 2019), it is valuable in microbial ecology. As for the alpha-diversity analysis, we chose to use the Hill numbers – which have several advantages (Roswell et al., 2021) – to provide an overview of fungal diversity within sites. This allowed us to observe all OTUs – "typical" OTUs and dominant ones –, based on their abundance frequencies (Chao et al., 2014), especially as fungal diversity can be represented by a few dominant species with a high relative abundance (Egidi et al., 2019).*

- *George, P. B., Creer, S., Griffiths, R. I., Emmett, B. A., Robinson, D. A., & Jones, D. L. (2019). Primer and database choice affect fungal functional but not biological diversity findings in a national soil survey. Frontiers in Environmental Science, 173.*

- *Nilsson, R. H., Tedersoo, L., Abarenkov, K., Ryberg, M., Kristiansson, E., Hartmann, M., ... & Kõljalg, U. (2012). Five simple guidelines for establishing basic authenticity and reliability of newly generated fungal ITS sequences. MycoKeys, 4, 37-63.*

- *Nilsson, R. H., Wurzbacher, C., Bahram, M., Coimbra, V. R., Larsson, E., Tedersoo, L., ... & Abarenkov, K. (2016). Top 50 most wanted fungi. MycoKeys, (12), 29-40.*

- *Banos, S., Lentendu, G., Kopf, A., Wubet, T., Glöckner, F. O., & Reich, M. (2018). A comprehensive fungi-specific 18S rRNA gene sequence primer toolkit suited for diverse research issues and sequencing platforms. BMC microbiology, 18, 1-15.*

- *Pauvert, C., Buée, M., Laval, V., Edel-Hermann, V., Fauchery, L., Gautier, A., ... & Vacher, C. (2019). Bioinformatics matters: The accuracy of plant and soil fungal community data is highly dependent on the metabarcoding pipeline. Fungal Ecology, 41, 23-33.*

- *Djemiel, C., Dequiedt, S., Karimi, B., Cottin, A., Girier, T., El Djoudi, Y., ... & Terrat, S. (2020). BIOCOM-PIPE: a new user-friendly metabarcoding pipeline for the characterization of microbial diversity from 16S, 18S and 23S rRNA gene amplicons. BMC bioinformatics, 21, 1-21.*

- *Terrat, S., Djemiel, C., Journay, C., Karimi, B., Dequiedt, S., Horrigue, W., ... & Ranjard, L. (2020). ReClustOR: a re-clustering tool using an open-reference method that improves operational taxonomic unit definition. Methods in Ecology and Evolution, 11(1), 168-180.*

- *Willis, A. D. (2019). Rarefaction, alpha diversity, and statistics. Frontiers in microbiology, 10, 2407.*

- *Castle, S. C., Samac, D. A., Sadowsky, M. J., Rosen, C. J., Gutknecht, J. L., & Kinkel, L. L. (2019). Impacts of sampling design on estimates of microbial community diversity and composition in agricultural soils. Microbial ecology, 78, 753-763.*

- *Roswell, M., Dushoff, J., & Winfree, R. (2021). A conceptual guide to measuring species diversity. Oikos, 130(3), 321-338.*

- *Chao, A., Gotelli, N. J., Hsieh, T. C., Sander, E. L., Ma, K. H., Colwell, R. K., & Ellison, A. M. (2014). Rarefaction and extrapolation with Hill numbers: a framework for*

*sampling and estimation in species diversity studies. Ecological monographs, 84(1), 45-67.*

- *Egidi, E., Delgado-Baquerizo, M., Plett, J. M., Wang, J., Eldridge, D. J., Bardgett, R. D., ... & Singh, B. K. (2019). A few Ascomycota taxa dominate soil fungal communities worldwide. Nature communications, 10(1), 2369.*

Abstract

L.40. Change "approach. Total cumulated" to "approach. The total accumulated".
We agree, we are going to change the sentence.

L.41. This "1%" is estimated based on which number? In a number mentioned above (ie. 12 million species), or on what? Try to be a bit more specific.
Yes indeed, we based ourselves on the most recent estimation of fungal diversity (Phukhamsakda et al., 2022). We completed the sentence.
*"Total cumulated fungal diversity across France included 136,219 OTUs, i.e., about 1% of the global soil fungal diversity (based on a maximum diversity estimate of 12 million) for a territory representing only 0.3% of terrestrial surface on Earth."*

L.49. Here, "distinct or similar filters" relative to what?
We agree, we completed the sentence by adding the distinct and similar drivers of the Hill numbers q=0 and q=2).
*The spatial distribution of abundant (q=2) and rare (q=0) fungi was determined by distinct (clay or organic carbon) or similar (soil pH) filters with various relative influences.*

Introduction

L.80. Not only nitrogen but also phosphorous.
We fully agree, thanks for pointing this out, we added phosphorous in the sentence.
*The functions of fungal communities provide many ecosystem services that promote mineral nutrition of plants linked to soil organic matter rates, phosphorous and nitrogen availability (Miyauchi et al., 2020; Ward et al., 2022).*

L.87. Change "agents and involved" to "agents and are involved".
We agree, we are going to change the sentence.

L.94-97. This is sort of true… but, check more recent, global sampling by the same author (Leho Tedersoo), as DOI: 10.1007/s13225-021-00493-7 and DOI: 10.1111/gcb.16398, among others.
We agree, we are going to add this reference study in the sentence.

L.130. Change "altitude) and climate" to "altitude), and climate".
We agree, we are going to change the sentence.

L.140. Change "types and soil characteristics" to "types, and soil characteristics of the country".
We agree, we are going to change the sentence.

L.141. How do you define pedological diversity?
We defined pedological diversity as the number of different soil types. The complete definition from the publication reads as follows, "The conventional pedodiversity measure is related to

the abundance of the observed soil type, thus there is a relationship between land area and Shannon's entropy." (Minasny, Budiman, Alex B. McBratney, and Alfred E. Hartemink. "Global pedodiversity, taxonomic distance, and the World Reference Base." Geoderma 155.3-4 (2010): 132-139.)

L.142. Change "(Minasny et al., 2010) and is also" to "(Minasny et al., 2010), and is also".
We agree, we are going to change the sentence.

L.144. Change "grasslands and forests" to "grasslands, and forests".
We agree, we are going to change the sentence.

L.146. Change "statements" to "facts".
We agree, we are going to change the sentence.

L.146-148. This is a very vague phrase… as a researcher based in Chile, I can say we also say this all the time… all countries are unique, thus, this phrase is not.
We fully agree, all countries are unique but the wide range of variation of climates, soils and land uses in France combined with an intensive and extensive soil sampling strategy leads us to conclude about the relevance of our study.
See our two sentences L146 to L151

L.149. Change "sampling strategy in the world and the systematic random" to "sampling strategies in the world, and this systematic random".
We agree, we are going to change the sentence.

L.154. Change "bioinformatics and statistical" to "bioinformatics, and statistical".
We agree, we are going to change the sentence.

L.156-157. At this point I have not read the rest of the manuscript… but it does make no sense at all to sequence the 18S to characterize soil fungal taxa. ITS1 or ITS2 are the regions to be sequenced.
We have detailed this response above, see the overall comment and the text that will be added to our discussion.

L.163. Change "diversity and inverse" to "diversity, and inverse". Change "prediction, (geostatistics)" to "prediction (geostatistics)".
We agree, we are going to change the sentence.

L.166. Change "conditions and land use" to "conditions, and land use".
We agree, we are going to change the sentence.

L.168. I do not get what you mean by "In fine".
Borrowed from Latin *in fine*, but we can change to "Finally".

Methods

L.178. Change "(n=65) and low anthropized" to "(n=65), and low anthropized".
We agree, we are going to change the sentence.

L.181. Change "were bulked" to "were combined".

We agree, we are going to change the sentence.

L.183. For how long these samples were air-dried? When too long, a bunch of fungal taxa might be lost. The correct approach is: either to dry the soil samples in oven (max. 40 C) or with silica gel, or freeze the samples (at -20 C or -80 C) less than 24 hours after collecting.
We agree, we will specify this point in the text.
*Each sample was air-dried following a standardized procedure at 35 °C until the soil humidity was below 1%, then sieved to 2 mm and separated into two sub-samples.*

L.185-187. Still, some brief description of soil chemical analyses is needed here.
We agree, we are going to change the sentence.
*A detailed description of the physicochemical analysis performed in this study (soil pH, texture, organic carbon, nitrogen, and phosphorus) is accessible from (Jolivet et al., 2006).*

L.195. Change "beads and 4 glass beads" to "beads, and 4 glass beads".
We agree, we are going to change the sentence.

L.197. Change "sulfate and up to" to "sulfate, and up to".
We agree, we are going to change the sentence.

L.209-211. Sadly, this selected regions is VERY inefficient to detect fungal diversity, ie. it only captures a very small fraction. It has happened to me.
We have detailed this response above, see the overall comment and the text that will be added to our discussion.
In addition, Banos et al. (2018) indicate that the regions we chose are most robust for characterizing fungal communities.

L.245. When you say "using the shapiro.test function", I know you refer to R, but you need to write of what package is that function (and correctly cite such package), and also, on which software -R- and cite such software.
We agree, we are going to clarify the sentence.
*We tested if the variables were normally distributed or approximately so, using the shapiro.test function (R "stats" package).*

L.247. Change "from forecast package" to "from the R forecast package".
We agree, we are going to change the sentence.

L.257. Change "as recommended by (Dini-Andreote et al., 2021)." to "as recommended by Dini-Andreote et al. (2021).", but, I am not sure what you mean by this.
We agree, we are going to clarify the sentence.
*The details of the samples removed (e.g., outliers) in the different land uses are available in Supplementary Figure 1, as recommended by Dini-Andreote et al. (2021) (Dini-Andreote et al., 2021), who developed a data management strategy with good practices for biogeographical studies.*

L.266. Change "the "usdm" package" to "the "usdm" R package".
We agree, we are going to change the sentence.

L.268. Change ""leaps" package" to ""leaps" R package".
We agree, we are going to change the sentence.

L.271. Change "functions ("vegan" package (Oksanen et al., 2013))." to "functions of the "vegan" R package Oksanen et al. (2013)).".
We agree, we are going to change the sentence.

L.272. Change "forward selection to" to "forward multiple regression selection to".
We agree, we are going to change the sentence.

L.277. Change "in (Granger et al., 2015)." to "in Granger et al. (2015).".
We agree, we are going to change the sentence.

L.279. Here, "using a local neighborhood" what?
Yes, we replaced the sentence by the following one:
Then, we predicted the unsampled locations by a global kriging approach that used all the points in the dataset (global neighborhood).

L.280. Change "the "gstat" package" to "the "gstat" R package".
We agree, we are going to change the sentence.

L.306. Cite this software correctly.
We agree, we are going to change the sentence.
*The networks were mapped using Cytoscape software (v. 3.9.1) (Shannon et al., 2003).*
*Shannon, P., Markiel, A., Ozier, O., Baliga, N. S., Wang, J. T., Ramage, D., ... & Ideker, T. (2003). Cytoscape: a software environment for integrated models of biomolecular interaction networks. Genome research, 13(11), 2498-2504.*

Results

L.328. Change "south-east and the" to "south-east, and the".
We agree, we are going to change the sentence.

L.340. Change "conditions and spatial" to "conditions, and spatial".
We agree, we are going to change the sentence.

L.345. Change "(0.39%) and spatial" to "(0.39%), and spatial".
We agree, we are going to change the sentence.

L.347. Change "(0.63%) and land use" to "(0.63%), and land use".
We agree, we are going to change the sentence.

L.350. Change "for q=1 and neared" to "for q=1, and neared".
We agree, we are going to change the sentence.

L.355. Change "was the pH" to "was pH".
We agree, we are going to change the sentence.

L.361, 363. Change "the soil pH" to "soil pH".
We agree, we are going to change the sentence.

L.364. Here, "were also identified" as what?

We agree, we are going to change the sentence.
*The soil pH remained the strongest driver with also a unimodal relationship, and organic carbon and the total iron content were also identified as secondary drivers, with a negative linear relationship.*

L.379. Change "soils and can be ranked" to "soils, and can be ranked".
We agree, we are going to change the sentence.

L.405. Change "with forest" to "with a pattern: forest".
We agree, we are going to change the sentence.

Discussion

L.420. Again, "1%" relative to what?
We agree, we are going to change the sentence.
Compared to the estimated worldwide diversity, France exhibits a very high cumulated
soil fungal richness (about 1% of the global soil fungal diversity based on a maximum diversity
estimate of 12 million) relative to its small surface (0.3 % of terrestrial land).

L.427. Change "were collected sites on" to "were collected in sites on".
We agree, we are going to change the sentence.

L.441. Change "hence possible biases" to "hence with possible biases".
We agree, we are going to change the sentence.

L.454. Change "OTUs), "typical"" to "OTUs), and "typical"".
We agree, we are going to change the sentence.

L.457. Change "radius and "spottier"" to "radius, and "spottier"".
We agree, we are going to change the sentence.

L.469. Change "the soil pH" to "soil pH". Change "by the clay content" to "by clay content".
We agree, we are going to change the sentence.

L.473. Change "of the pH in the" to "of soil pH in the".
We agree, we are going to change the sentence.

L.475. Change "the soil pH" to "soil pH".
We agree, we are going to change the sentence.

L.496-498. The last part of this phrase ("whatever the Hill number") is not clear.
We removed the « whatever the Hill number (Terrat et al., 2017).
To clarify the text

L.516. Change "2017) and fungi" to "2017), and fungi".
We agree, we are going to change the sentence.

L.517. Change "grassland and crop" to "grassland, and crop".

We agree, we are going to change the sentence.

L.534. Change "soil structure and biological" to "soil structure, and biological".
We agree, we are going to change the sentence.

Conclusions

L.584. Change "processes and environmental" to "processes, and environmental".
We agree, we are going to change the sentence.

Reviewer 2:

The manuscript addresses soil fungal alpha diversity and network analysis based metabarcoding using a comprehensive and extensive sampling of over 2,000 sites across France. This sampling grid provides an enormous wealth of data on soil fungi. Effect of land-uses and climatic regions as well as physico-chemical soil factors on different alpha-diversity measures were assessed.

**My main concerns are about some methodological issues and the conciseness and relevance of the discussion. Following methodological questions came up:**

What is the advantage of including Hill numbers based on three different orders of diversity?

We chose to include Hill numbers based on three different orders of diversity because they have been widely used in microbial ecology in recent years to describe alpha diversity. Behind these Hill numbers are traditional indices used in ecology, that is to say:
- q=0: richness
- q=1: the Shannon index, calculated using the exponential
- q=2: the inverse Simpson index.
Therefore, only the Hill numbers (q=1) differed from the traditionally used indices.
Regarding the benefits of using these Hill numbers, the review by Roswell et al. (2021) describes this very well. The three main advantages are:
- Hill diversities behave in ways that are logically reasonable for a measure of diversity (Hurlbert 1971, Jost 2009, Tuomisto 2010).
- the calculation of Hill diversity is simple and already familiar to ecologists. Like the other traditional diversity indices, the Hill diversity index summarizes relative (but not absolute) abundances, and the only data required to compute the Hill diversity of a sample are the relative abundances of species in a sample.
- the authors mentioned below develop the idea that rarity can be defined as the reciprocal of relative abundance, and that Hill diversity indices calculate the mean of the rarities of the species in the sample.
Roswell, Michael, Jonathan Dushoff, and Rachael Winfree. "A conceptual guide to measuring species diversity." Oikos 130.3 (2021): 321-338.

What is the relationship between the Hill numbers of q=0, q=1 and q=2 in your dataset? Do they correlate? What is the biological difference between Hill numbers based on q=2 and q=3? Or in other words what is the justification to include Hill numbers based on q=2?
The relationships between the Hill numbers based on q=0, q=1 and q=2 in our dataset are as follows (see figure):

- a significant strong positive relationship between q=0 and q=1 and between q=1 and q=2
- a significant positive relationship between q=0 and q=2.

[Figure]

The Hill numbers based on q=3 can indeed be calculated, but we did not do so because to our knowledge they are rarely used in microbial ecology. The Hill numbers based on q=2 correspond to the inverse Simpson index, traditionally used because it gives more weight to common or dominant species. In that way, the few rare species with only a few representatives in a sample will not affect this diversity index.

We will provide more details in the answers to the questions below. However, we based ourselves on the various publications by Anne Chao who worked on diversity indices and the concept of Hill numbers for a very long time.

Despite a correlation between these indices, they were complementary for understanding the ecology of soil fungal communities (e.g., mapping, environmental filters).

Chao, A. et al. 2014. Rarefaction and extrapolation with Hill numbers: a framework for sampling and estimation in species diversity studies. – Ecol. Monogr. 84: 45–67.

Chao, Anne, Chun-Huo Chiu, and Lou Jost. "Unifying species diversity, phylogenetic diversity, functional diversity, and related similarity and differentiation measures through Hill numbers." Annual review of ecology, evolution, and systematics 45 (2014): 297-324.

Chiu, Chun-Huo, and Anne Chao. "Distance-based functional diversity measures and their decomposition: a framework based on Hill numbers." PloS one 9.7 (2014): e100014.

What was the rational to use Hill numbers based on q=3 to define the abundant fungal taxa. Often the abundant community is defined as OTUs with a relative abundance greater than some threshold (e.g., 0.1%) of the total sequence abundance. This will select very different taxa. In the discussion, results were compared to studies with the latter definition of abundant fungi. Please explain.

We checked the entire manuscript: we do not use the Hill numbers based on q=3.

We also think that it is preferable to keep all the OTUs (sequencing now provides very high-quality reads, and the various filters make it possible to keep OTUs consistent) and to calculate

indices to observe alpha diversity (i.e., in a sample) using Hill numbers rather than an abundance threshold that seems very arbitrary.

We agree that we often discuss our results relatively to publications using thresholds because very few studies in soil microbial biogeography use the different diversity indices. We hope that more and more authors will take Hill numbers into account to explore alpha diversity in microbial ecology.

What was the rational of assessing alpha-diversity and network analyses, but omitting many other community measures such as beta diversity?

In this paper, which is focused on alpha diversity and interaction networks, we chose to focus on a deep and comprehensive analysis of diversity within samples independently. Concerning beta diversity analysis, this will be the subject of another paper also including a complete analysis with the structure of communities and the characterization of fungal habitats.

Very difficult to assess the networks in the way they are presented by eye. How did you deal with uneven number of sites (between 65 to 886 sites)? Why did you consider networks for land uses but not for climatic regions?
We understand the reviewer's comment, and we improved the way we present these results to the readers of the journal. We added the metrics of the network analysis directly in the manuscript (for land uses and climates). We also added the analysis of the networks under different climates directly in the manuscript. We added color to the figures to indicate whether the interaction is positive or negative (also known as co-occurrence and co-exclusion, respectively). We added information about the addition of climate networks to the Methods section. As for the methodology, we compared the networks with each other based on Karimi et al. (2019; 2020), as indicated L291-L294.

Comparing the fungal networks of the four land uses or under the eight climates required: i) standardizing the number of soils used to compute the network *per* land use to avoid a sampling size effect, and ii) network replicates that integrated natural soil heterogeneity within each land use. As vineyards & orchards provided only 65 soil samples at the territorial scale and the altered Mediterranean climate (Type 6) provided only 86 soil samples, we standardized the sampling size at 60 sites *per* network for all land uses and at 83 for all climates, drawn from the respective pools of soils. Thus, the minimum number of combinations (8,259,888 for land uses and 102,340 for the climate analysis) ensured that each network was computed from a unique combination of sites. The number of replicates used to evaluate the range of variation within and between land uses was set at 100. The 60 or 83 samples were drawn independently for each of the 100 network replicates. Digital sampling of the sites was monitored. Among the 1,982 sites classified as forest, grassland, crops, or vineyards & orchards, 1,978 were sampled at least once, and among the 2,057 sites classified for climate, 2,057 were sampled at least once. This replication procedure ensured that more than 99.80% of the land uses and all the climates of the available sites were sampled for a standardized analysis of all land uses or climates, while being representative of French soils in general.

In the Introduction section, we are going to change the text:

*In fine*, we compared the variations of fungal diversity across different land uses plus climate types with the variations of the complexity of the fungal interaction networks by inferring co-occurrence networks at the operational taxonomic unit (OTU) level.

In the Methods section, we are going to add the following text in '2.1. Soil sampling design':
… and eight climatic regions: type 1 (n=284), type 2 (n=270), type 3 (n=545), type 4 (n=418), type 5 (n=265), type 6 (n=86), type 7 (n=94), and type 8 (n=95).

In the Methods section, we are going to change the text of '2.5. Fungal co-occurrence networks':

[revised manuscript text omitted]

**The discussion could be improved by addressing minor issues listed below.**

Focus on soil fungi, including other soil organisms seems off-topic (line 428-432). Particularly, the term "soil microorganisms" includes many different types of organisms with very different ecologies and life styles, therefore I would prefer to put the focus of this work only on soil fungi and to avoid confusions omit the term "soil microorganisms" (line 448, 491, 495, 535, …).

We agree, we are going to remove the sentences from L428 to L432. In L448, we removed Fuhrman, 2009; Pedrós-Alió, 2012; Bickel and Or, 2021 and only kept Egidi et al., 2019.
We are going to change the sentence.
The soil fungi community is well known to be largely dominated by a few highly abundant taxa and to include a large number of rare taxa (Egidi et al., 2019).
L491 We are going to change the sentence.
These metallic elements occur naturally but also result from human activities and are known to be toxic for soil fungi when accumulated in the environment (Sun et al., 2022; Ding et al., 2022).
L495 We are going to change the sentence.
… and our results could reflect the significant impact of this contamination on soil fungi at a broad scale.
L535 We are going to change the sentence.
… and could favor the development of soil fungi, as previously described at the landscape scale in Brittany (western France, Le Guillou et al., 2019).

It was stated that abundant species are generalist and copiotrophs (line 464-465, 470-472, 503-506). First, abundant species were defined here as Hill numbers based on q=2, which is different than how it was in these references (see comment above). Therefore, this statement seems speculative. Second, you could investigate based on taxonomy of your data, if the abundant taxa are indeed more likely to be copiotrophs or generalists.
Please see our previous comments about the Hill numbers *vs.* relative abundance thresholds. However, we agree, and we will modify the text to mitigate our remarks by adding 'putative'. Concerning the taxonomic analysis, this will be the subject of a future paper also including a complete analysis, fungal taxonomy and ecological traits.

Why did you chose the v7-v8 of the 18S instead of ITS which is considered the standard marker for fungal communities. How do the markers differ? Do you expect higher or lower diversity? How did you make sure that only fungal sequences were included in the dataset?
Reviewer 1 also highlighted the choice of 18S *vs.* ITS, so we are making a joint response.
We agree with the reviewer that ITS is the most common and historical molecular marker for assessing fungal taxonomy, especially to describe taxonomic groups. However, 18S rDNA has also been an alternative in the last few years to assess the soil fungal community structure and richness more globally, as demonstrated by several authors – in particular in biogeography and at large scales (see below).

- George, P. B., Creer, S., Griffiths, R. I., Emmett, B. A., Robinson, D. A., & Jones, D. L. (2019). Primer and database choice affect fungal functional but not biological diversity findings in a national soil survey. Frontiers in Environmental Science, 173.

- Li, J., Delgado-Baquerizo, M., Wang, J. T., Hu, H. W., Cai, Z. J., Zhu, Y. N., & Singh, B. K. (2019). Fungal richness contributes to multifunctionality in boreal forest soil. Soil Biology and Biochemistry, 136, 107526.

- Bastida, F., Eldridge, D. J., García, C., Kenny Png, G., Bardgett, R. D., & Delgado-Baquerizo, M. (2021). Soil microbial diversity–biomass relationships are driven by soil carbon content across global biomes. The ISME Journal, 15(7), 2081-2091.

- Fernandes, M. L. P., Bastida, F., Jehmlich, N., Martinović, T., Větrovský, T., Baldrian, P., ... & Starke, R. (2022). Functional soil mycobiome across ecosystems. *Journal of Proteomics*, *252*, 104428.

We will add a specific paragraph in the discussion of the manuscript justifying the relevance of our molecular marker, and more broadly about the metabarcoding approach (L432-L433).

***Relevance and drawbacks of the metabarcoding approach***
*Two molecular markers are commonly used to explore fungal diversity thanks to metabarcoding approaches: the internal transcribed spacer (ITS) region, accepted as a universal barcode, and the 18S rRNA gene as an alternative. Both have advantages and drawbacks, in particular to observe specific functional groups. For example, members of the class Glomeromycetes are better characterized using 18S rDNA than ITS, especially in the soil microbiota, and this could have a significant impact on fungal diversity metrics (George et al., 2019). In addition, important reference sequences are only annotated at the phylum level in the international databases (Nilsson et al., 2012; Nilsson et al., 2016; Banos et al., 2018). Moreover, a recent study highlighted that national-scale fungal biogeography studies based on 18S rDNA were relevant to decipher the relationships between fungal diversity and environmental filters (George et al., 2019). For all these reasons, we chose to use the 18S rDNA gene to characterize fungal diversity. Once the molecular marker was chosen, the hypervariable region had to be selected for sequencing. Various criteria had to be taken into account such as amplicon length in relation to the sequencer, or the desired taxonomic and phylogenetic resolution. We sequenced the V7-V8 regions because they appeared to be the most promising regions for fungal diversity assessment (Banos et al., 2018). The last tricky step of a fungal diversity study is the bioinformatics analysis, which is dependent on the pipeline (Pauvert et al., 2019). Nevertheless, our previous studies on bacterial biogeography showed that our metabarcoding pipeline and associated tools were highly appropriate to analyze large microbial datasets (Djemiel et al., 2020; Terrat et al., 2019).*
*Although alpha-diversity depends on the intensity and strategy of sampling, especially when addressing the complexity of the soil microbiota (Willis et al., 2019; Castle et al., 2019), it is valuable in microbial ecology. As for the alpha-diversity analysis, we chose to use the Hill numbers – which have several advantages (Roswell et al., 2021) – to provide an overview of fungal diversity within sites. This allowed us to observe all OTUs – "typical" OTUs and dominant ones –, based on their abundance frequencies (Chao et al., 2014), especially as fungal diversity can be represented by a few dominant species with a high relative abundance (Egidi et al., 2019).*

- *George, P. B., Creer, S., Griffiths, R. I., Emmett, B. A., Robinson, D. A., & Jones, D. L. (2019). Primer and database choice affect fungal functional but not biological diversity findings in a national soil survey. Frontiers in Environmental Science, 173.*
- *Nilsson, R. H., Tedersoo, L., Abarenkov, K., Ryberg, M., Kristiansson, E., Hartmann, M., ... & Kõljalg, U. (2012). Five simple guidelines for establishing basic authenticity and reliability of newly generated fungal ITS sequences. MycoKeys, 4, 37-63.*
- *Nilsson, R. H., Wurzbacher, C., Bahram, M., Coimbra, V. R., Larsson, E., Tedersoo, L., ... & Abarenkov, K. (2016). Top 50 most wanted fungi. MycoKeys, (12), 29-40.*
- *Banos, S., Lentendu, G., Kopf, A., Wubet, T., Glöckner, F. O., & Reich, M. (2018). A comprehensive fungi-specific 18S rRNA gene sequence primer toolkit suited for diverse research issues and sequencing platforms. BMC microbiology, 18, 1-15.*
- *Pauvert, C., Buée, M., Laval, V., Edel-Hermann, V., Fauchery, L., Gautier, A., ... & Vacher, C. (2019). Bioinformatics matters: The accuracy of plant and soil fungal*

*community data is highly dependent on the metabarcoding pipeline. Fungal Ecology, 41, 23-33.*

- *Djemiel, C., Dequiedt, S., Karimi, B., Cottin, A., Girier, T., El Djoudi, Y., ... & Terrat, S. (2020). BIOCOM-PIPE: a new user-friendly metabarcoding pipeline for the characterization of microbial diversity from 16S, 18S and 23S rRNA gene amplicons. BMC bioinformatics, 21, 1-21.*
- *Terrat, S., Djemiel, C., Journay, C., Karimi, B., Dequiedt, S., Horrigue, W., ... & Ranjard, L. (2020). ReClustOR: a re-clustering tool using an open-reference method that improves operational taxonomic unit definition. Methods in Ecology and Evolution, 11(1), 168-180.*
- *Willis, A. D. (2019). Rarefaction, alpha diversity, and statistics. Frontiers in microbiology, 10, 2407.*
- *Castle, S. C., Samac, D. A., Sadowsky, M. J., Rosen, C. J., Gutknecht, J. L., & Kinkel, L. L. (2019). Impacts of sampling design on estimates of microbial community diversity and composition in agricultural soils. Microbial ecology, 78, 753-763.*
- *Roswell, M., Dushoff, J., & Winfree, R. (2021). A conceptual guide to measuring species diversity. Oikos, 130(3), 321-338.*
- *Chao, A., Gotelli, N. J., Hsieh, T. C., Sander, E. L., Ma, K. H., Colwell, R. K., & Ellison, A. M. (2014). Rarefaction and extrapolation with Hill numbers: a framework for sampling and estimation in species diversity studies. Ecological monographs, 84(1), 45-67.*
- *Egidi, E., Delgado-Baquerizo, M., Plett, J. M., Wang, J., Eldridge, D. J., Bardgett, R. D., ... & Singh, B. K. (2019). A few Ascomycota taxa dominate soil fungal communities worldwide. Nature communications, 10(1), 2369.*

Whether diversity is higher when studying 18S rRNA or ITS is a difficult question because it is dependent on many parameters such as the choice of the targeted region, the PCR conditions, the sequencer, the bioinformatics analysis (quality filters, rare sequence filtering, clustering threshold, and subsampling). Overall, we expect to observe more OTUs when using 18S rRNA than when using ITS, as in George et al. (2019).

Finally, to make sure that we only keep fungal sequences, a step in our BIOCOM-PIPE pipeline (Djemiel et al., 2020) checks the taxonomic assignment of our sequences against a database exclusively containing fungal sequences. (SILVA).
We are going to add the following sentence:
An additional step allowed us to check whether all our sequences were indeed affiliated to fungal sequences.

Point-by-point responses

Line 36: How does your work contribute to the description of soil fungi?
The overall objectives of this study were to i) describe the fungal diversity variations across France, and ii) identify the environmental filters of this diversity (soil properties, land management, climate, spatial descriptors). Thanks to our intensive soil sampling strategy combined with metabarcoding based on high-throughput sequencing, we think that our work contributes to the description of soil fungi.

Line 49-50: Sentence with very little information. Rephrase or omit.
We agree, we are going to change the sentence.

As related to other environmental parameters, the spatial distribution of fungal diversity (Hill numbers based on different orders of diversity) was mainly influenced by local filters such as soil characteristics and land management, but also by global filters such as climate conditions with various relative influences.

Line 58: "global change" was not a topic assessed in this work.
We agree, we are going to remove the end of the sentence.
Our findings provide novel insights for a better understanding of soil fungal ecology and upgrade biodiversity conservation policies by supplying representative repositories dedicated to soil fungi.

Line 80: What is the "soil organic matter rate" in this context?
We agree, we are going to replace 'Rate' by 'turnover' in this sentence.

Line 88: There are fungal species that are used as biocontrol agents not only against pathogenic microorganisms but also against insects (entomopathogenic fungi).
We agree, we are going to change the sentence.
Some of them are also identified as biocontrol agents and involved in plant protection through the regulation of pathogenic microorganisms and insect pests (Peng et al., 2021).

Line 131: "… yet to a lesser extent." compared to what? Please clarify.
We agree, we are going to remove 'yet to a lesser extent' to clarify this sentence.

Line 162: Hill number based on order of diversity of two is the exponential of the Shannon diversity. Please correct.
We agree, we are going to change the sentence.
To reach this objective, we used Hill numbers to combine complementary diversity indexes such as richness, exponential Shannon diversity and inverse Simpson.

Line 156-171: This paragraph should be improved by stating the objectives more clearly. It is more focused on what you did to answer the objective.
We agree, we are going to change the first sentence.
We explored the spatial distribution of the soil fungal communities on a broad geographical scale in order to better understand fungal ecology according to environmental filters using high-throughput sequencing of the small 18S rDNA gene subunit directly amplified from soil DNA.

Line 167: Which processes did you quantify and how?
We agree, we are going to remove 'the ecological processes' and clarify this sentence.

Line 178: What means "Low anthropized" in France?
We are going to add this sentence to clarify.
In France, low anthropized environments group wetlands, peatlands, sclerophyllous forests, natural grasslands, sparsely vegetated areas, and bare rocks (Karimi et al., 2020).

Line 181: "bulked" is not a common English word. Rather use "combined" or "homogenized".
We agree, we are going to change the sentence to clarify.
The core samples were pooled and homogenized to obtain a composite sample.

Line 203: Why did you switch from "DNA" singular to "DNAs" in the plural form?
We agree, we are going to change all sentences in the manuscript with 'DNA'.

Line 206: Did you mean "5 ng / µl"?
We agree we are going to change the sentence to clarify.

Line 228: "The libraries were demultiplexed and trimmed with zero difference between the barcode and primer sequences." If I understand this sentence correctly instead of "between" you should use "to", because you are not comparing barcode and primer, but barcode and primer to the OTU consensus sequence.
To clarify this point, we chose to check each read from the entire library to find the correct sequence for the barcode. If one correct barcode sequence was detected, the read was checked to find the complete and perfect primer sequence. If the primer sequence was also perfectly detected, we kept the read for further analyses, more specifically chimera checking. If a difference was found in the barcode sequence or in the primer sequence, the read was deleted from the dataset.

Line 231: Why OTUs based on 95% identity?
The analysis of microbial community diversity relies on the construction of similarity clusters (called OTUs) of rRNA gene PCR amplicons (Horner-Devine et al., 2004). We chose to use OTUs to examine the distribution of 18S rRNA gene sequences in our datasets. However, there is no single best definition of 'species', 'genus', etc. when this approach is used, because of a controversy about thresholds of similarity allowing clear differentiation of taxonomic units. Moreover, with short sequences such as those obtained with the Illumina technology used in this study, the best taxonomic resolution was the genus level. In previous studies on microbial eukaryotes encompassing fungi (Logares et al., 2014 – Current Biology), the authors chose a threshold of 95% to minimize any inflation of diversity estimates caused by remaining errors (Kunin et al., 2010 – Environmental Microbiology). In addition, they tested a 97% OTU clustering threshold, and comparable patterns of the proportions of locally abundant and rare subcommunities were obtained (Logares et al., 2014 – Current Biology). This is in line with other studies working on 18S and/or ITS (Botnen et al., 2018 – Molecular Ecology Resources), showing that the resulting community structures were highly similar across all clustering thresholds, explained by the existence of strong ecological structuring gradients and phylogenetically diverse sets of abundant OTUs highly stable across clustering thresholds. Moreover, to confirm the efficiency of the 95% threshold for the part of the 18S rRNA gene we chose to target, we performed an *in-silico* approach (data not showed). We first extracted all known and reliable fungal sequences from the SILVA database (release 114). We only kept the regions of the sequences amplified by our primer set, and deleted those with too many mismatches with our primer set (> 3 mismatches with one primer). Then, all these "artificial reads" with a reliable taxonomy were clustered using our pipeline at various threshold levels (100% to 90%, with a 0.1% step). The 95% threshold was best suited to efficiently define the genus level based on the results and the amplified regions, as it was closest to the genus level defined by the SILVA taxonomy of the sequences.

Line 236: Please include references for this statement.
We agree, we are going to add the reference.
… described in the literature (Terrat et al., 2017; Karimi et al., 2019).

Line 241-242: I would say that OTU richness does not emphasize any OTU but treats all OTUs equally, independent of the abundance.

In the richness analysis, all OTUs are indeed treated equally and independently of their abundance, and correspond to the count of the presence of OTUs in the sample. However, the distribution of sequences within OTUs is very variable, and most of the OTUs within a sample are only represented by a few sequences (see figure). We can consider that OTUs with one, two or three sequences are rare OTUs, and this cumulatively represented 81% of total richness in our dataset (see fig).

[Figure]

Hill numbers are a parametric family of diversity indexes differing among each other only by parameter q that determines sensitivity to OTU relative abundances (see previous comments). Therefore, Hill numbers are equally sensitive to rare and abundant OTUs; sensitivity to rare species increases as q decreases, and sensitivity to abundant species increases as q increases (Jost, 2007; Morris et al., 2014). Therefore, the richness index is sensitive to rare OTUs, q1 and q2 are sensitive to abundant OTUs.

Morris, E. Kathryn, et al. "Choosing and using diversity indices: insights for ecological applications from the German Biodiversity Exploratories." *Ecology and evolution* 4.18 (2014): 3514-3524.
Jost, Lou. "Partitioning diversity into independent alpha and beta components." Ecology 88.10 (2007): 2427-2439.

Line 243: The word "common" means for me widely-distributed, i.e., in different samples, therefore it does not fit to describe Hill diversity of order 1.

We based ourselves on the literature of Anne Chao, who worked extensively on the subject, and we kept her terminology. For example, according to Chao et al. (2010), 'Roughly, 1D measures the number of 'common' (or 'typical') species in a community.'

Chao, Anne, Chun-Huo Chiu, and Lou Jost. "Phylogenetic diversity measures based on Hill numbers." Philosophical Transactions of the Royal Society B: Biological Sciences 365.1558 (2010): 3599-3609.

Line 257: What was recommended by the authors of the article from Dini-Andreote et al., 2021.
We agree, we are going to change the sentence to clarify.
The details of the removed samples (i.e., the samples considered to be outliers and the 'low anthropized environment' samples that we decided to exclude) in the different land uses are available in Supplementary Figure 1, as recommended by Dini-Andreote et al. (2021).

Line 261: Supplementary Figure 2 and 3. Please specify, "for all these reasons". What are these reasons?
We agree, we are going to change the sentence to clarify.
Therefore, we performed a polynomial transformation of degree 2 for the soil pH variable to improve the accuracy of fitting for the variance partitioning analysis.

Line 277: Parentheses should be placed around the year of publication. (occurs at several places throughout the manuscript).
We agree, but we can't change the references ourselves because we used the journal format with Word and Zotero, which impose automatic formatting. Perhaps the journal can do this during proofs and we will identify the references to be changed at that moment.

Line 290: Parentheses should be placed around the year of publication.
Please see previous comment.

Line 300: Why did you use 0.06 as a cut-off for p-value?
We set the p-value at 0.06 because during the p-value adjustment step to check the robustness of the correlation between taxa, we performed a very stringent correction using the false-discovery rate method (method of Benjamini, Hochberg a.k.a. BH).

Line 303: Parentheses should be placed around the year of publication.
Please see previous comment.

Line 304-305: To improve clarity, the sentence should be rewritten to something like "comparisons of the fungal network metrics between land uses".
We agree, we are going to change the sentence to clarify.
We used a Kruskal-Wallis test with Bonferroni correction for the multiple comparisons of the fungal networks between the land uses and climate types, using 100 repetitions.

Line 305: Which of the 100 repeated networks was mapped? How much did they differ?
We represented the median network. The differences can only be observed at the level of the network metrics. We are going to change the sentence to clarify.
The median networks were mapped using Cytoscape version 3.9.1.
Line 311-313: The word validate does not fit in the way it was used here. Replace by something like: "The use of bioinformatic filters resulted in 2,060 samples with sufficient high-quality data for the assessment of fungal communities."
We agree, we are going to change the sentence to clarify.

The use of bioinformatic filters (in BIOCOM-PIPE workflow) resulted in 2,060 samples with sufficient high-quality data for the assessment of fungal communities.

Line 314-316: Unlogic sentence. The sentence "Thanks to our intensive soil sampling strategy combined with in-depth sequencing, we extrapolated with iNEXT a total of 186,794 OTUs (Fig. 1) at the national scale." should be rewritten to something like "… we obtained an extrapolated total value of 186,794 OTUs." It should be acknowledged that the curve is not near to flattening and that this suggests an even higher total diversity. Additionally, in Fig.1 the x-axis displays "sequences" rather than "individuals".
We agree, we are going to change the sentence to clarify.
Thanks to our intensive soil sampling strategy combined with in-depth sequencing, we obtained an extrapolated total value of 186,794 fungal OTUs (Fig.1) at the national scale.

Line 319: "We generated three national maps showing the soil fungal alpha-diversity for all Hill numbers usi ng a kriging interpolation approach (Fig. 2)." Preciseness of this sentence should be improved, e.g., "… soil fungal alpha-diversity based on Hill numbers with orders of diversity of zero, one and two …". As I understand it, one advantage of the Hill numbers is a better comparability among measures. In order to compare the three maps, the colors need to be united. The range of Hill numbers differed for the orders of diversity, i.e., order of zero resulted in a range of 1300 to 1600, order of 1 resulted in a range of 99 to 164 and order of two resulted in a range of 15 to 28 but the color ranges are always the same (from blue to read for minimum and maximum values). Therefore, a large change in color in Figure 2c is based on a change of just a few dominant OTUs and may falsely reveal a spotty distribution.
We agree, we are going to change the sentence to clarify.
We generated three national maps showing the soil fungal alpha-diversity based on Hill numbers with orders of diversity of zero, one and two using a kriging interpolation approach (Fig. 2).
Concerning the second part of the comment, we cannot put the three maps on the same color scale because the orders of ranges of the Hill numbers are different. If we do this, the maps will be homogenous, and we will not see any difference. Generating three maps based on Hill numbers makes it possible to precisely observe the variations in fungal diversity between rare and dominant OTUs. One region may have high richness and a low number of dominant OTUs, and another region may have lower richness but a high number of dominant OTUs.

Line 332-335: What does this mean from an ecological perspective? Further, the difference between more homogenous vs heterogenous coloring among the maps is probably due to the scale of the coloring. In map of Figure 2a, a change in a few species is not visible.
We discuss this in the Discussion section (L448-L461). However, we are going to add a sentence in the text to clarify.
These results altogether suggest a 'patchier' pattern, generally considered less stochastic than a 'spottier' distribution (Dequiet et al., 2011; Terrat al., 2015).
Dequiedt, Samuel, et al. "Biogeographical patterns of soil molecular microbial biomass as influenced by soil characteristics and management." Global Ecology and Biogeography 20.4 (2011): 641-652.
Terrat, Sebastien, et al. "Improving soil bacterial taxa–area relationships assessment using DNA meta-barcoding." Heredity 114.5 (2015): 468-475.

Concerning the scale of the coloring, please see previous and next comments.

Line 340: It is not "different Hill numbers" but "Hill numbers based on different orders of diversity". Check this phrase in the entire document.
We agree, we are going to change all sentences to clarify.

Line 350: Change to "4.25" instead of "4,25".
We agree, we are going to change the sentence.

Line 350-351: Are these results discussed? Either explain the most important findings or delete.
We agree, we are going to remove the sentence.

Line 355: What is this number "6.72%"? From which analyses and data does it come from?
We agree, we are going to change the sentence to clarify.
The key determining soil parameter for fungal richness was the pH (variation explained by 6.72%).

Line 360: I do not see available phosphorus in Figure 3b and what about total nickel and cadmium?
We agree, it is an error; we are going to change the sentence.
Conversely, a positive relationship was observed with the total lead content (0.58%) and a negative relationship with the total nickel content (0.53%) and the total cadmium content (0.37%) (Fig. 3b).

Line 361: A "comparable distribution" to what?
We agree, we are going to change the sentence by removing 'comparable'.

Line 364: What about silt?
We agree, we are going to add a sentence.
Moreover, a weak positive relationship was observed with silt (variation explained by 0.16% with a significance level of 0.1)

Line 367-374: Please reconsider and rewrite carefully, some results are not significant. For instance, the Hill numbers of order of diversity zero and one in type 1 do not differ significantly from type 6, 7 and 8.
We agree, we are going to add a sentence to clarify.
Moreover, we observed no significant difference from types 6, 7 and 8.

Line 373: Please, explain why you think that this is interesting. What does it mean from an ecological perspective that the results differ for the Hill numbers based on different orders of diversity?
We agree, we are going to change the sentence to clarify.
Regarding the dominant OTUs (q=2), there was no significant difference in fungal diversity across the various climate types of France (Fig. 6d).
The meaning of the different Hill numbers based on different orders of diversity from an ecological perspective is discussed in the Discussion section (L496-L509).

Line 380: Does the x represent the mean? In the text, rather use the word "mean", because x could be confusing and unclear for the reader as it was not defined.
In statistics, the calculated mean is usually noted x using the macron (diacritic) x̄. There may have been a problem generating the PDF.

Line 383-384: It is unclear what you mean with this sentence. "Furthermore, the extreme fungal diversity values greatly varied according to the land use, whatever the metrics used."
We agree, we are going to remove the sentence to clarify.

Line 393: For clarity, add "compared to crops without grassland rotation".
We agree, we are going to change the sentence to clarify.

Significant differences were also recorded by comparing the different land managements of the crop systems: whatever the metrics, soil fungal diversity was higher under crops with a grassland rotation than under crops without a grassland rotation.

Line 397-402: It is not possible to assess these networks by eye, because they are so small and tight. Instead or in addition of displaying the networks, the network metrics (Supplementary Figure 5) should be displayed in the main document.
We agree, we are going to improve the network analysis (see previous comment about networks across climate types). We moved the network metrics (Supplementary Figure 5) to the main document and added the ratio between the positive and negative links. We also displayed the positive (green) and negative (red) links in the network visualization. We will also add information in the Methods section, and we discuss these results in the Results and Discussion sections.

Line 413 & 415: Be consistent with comma and points for decimal separators: 2.2, 3.8 and 6.28.
We agree, we are going to use points as decimal separators.

Line 420: Where does this number of 1% come from? I miss a discussion about extrapolations, the lack of flattening of the curve and differences in resolution of 18S vs ITS marker.
Reviewer 1 also highlighted this number of 1% and the choice of 18S *vs*. ITS, so we are making a joint response.
Yes indeed, we based ourselves on the most recent estimation of fungal diversity (Phukhamsakda et al., 2022). We completed the sentence.
L.41. "Total cumulated fungal diversity across France included 136,219 OTUs, i.e., about 1% of worldwide soil fungal diversity (based on a maximum diversity estimate of 12 million) for a territory representing only 0.3% of terrestrial surface on Earth."
L.420. Compared to the estimated worldwide diversity, France exhibits a very high cumulated soil fungal richness (about 1% of the global soil fungal diversity based on a maximum diversity estimate of 12 million) relative to its small surface (0.3 % of terrestrial land).

Line 421: I do not fully agree with the argument "that global soil fungal diversity is strongly under-estimated worldwide", because the identity of the fungi is not taken into consideration. Possibly there are many wide-spread species, so the total richness will not be additive if larger areas are sampled.
We agree, we are going to tone down the sentence by removing 'strongly', and replace 'diversity' by 'richness'. However, we studied the identity of the fungi through an OTU approach, not through a taxonomic approach. In this sentence, we only talk about richness. We also agree that total richness may not be additive if larger areas are sampled (we don't know).
This suggests that global soil fungal richness is under-estimated worldwide mainly due to the poor intensive sampling strategy that has only been extensive to date, with few sampling sites.

Line 429-432: Why do you compare here to these selected groups of soil organisms? This requires an explanation and what you conclude from that.
We agree, we are going to remove the sentence to clarify.

Line 435-440: I do not think that this is a fair comparison. How many of the samples in the global study were in France? Probably just a few.
We wanted to highlight that the strategy for capturing heterogeneity has to be adapted, in particular for national scales because global biogeography studies based on only a few or no sampling points across countries very poorly extrapolate to these non-sampled geographic areas.

Line 445-446: I do not agree with the wording used to describe the Hill numbers based on order of diversity of one and two, i.e., "including …". Rather appropriate is the term "weighing".

From Alberdi & Gilbert (2019):

"A q value of 1 (in practical terms its limit, as the Hill number is undefined for q = 1) is the value that weighs OTUs by their frequency, without disproportionately favouring either rare or abundant ones (Jost, 2006)." And "When a q value of 2 is applied, abundant OTUs are overweighed …"

Also see comment from line 243 about "common".

We think you're talking about lines 454-456. We agree, we are going to replace with 'weighting' to clarify. We also use the term 'weighting' L323.

Also, what is the difference between "spotty" and "patchy". For me both mean the same and "patchy" is more commonly used in ecology.
Please see L454-L458 in the manuscript … "patchier" (i.e., spatially more diffuse) for q=0 with 231 km radius and "spottier" (i.e., spatially more restricted) for q=1 (27 km) and q=2 (36 km).

Also, see comment from line 319. I am missing a discussion what the differences of the spatial distribution of the Hill numbers of different orders or diversity mean for the ecology of fungi, if they still exist after adapting the coloring to the same scale.
Please see previous comment in the Results section.

Line 458-461: I question your definition of abundant taxa. Hill numbers of order of diversity two, reveals the number of dominant taxa in each sample, but they could be very different OTUs in different samples. This numbers are about the number of different dominant OTUs in a sample, so about the shape of a community profile per sample and how it changes across the country. In the article that you compared your work with dominant and rare was defined across the entire dataset as follows:

Jiao and Lu (2020):

"Here, OTUs with relative abundances above 0.1% of the total sequences were defined as "abundant" taxa, those with relative abundances below 0.01% were defined as "rare" taxa, and those with relative abundances between 0.01% and 0.1% were "intermediate" OTUs (Jiao, Chen, et al., 2017; Liu et al., 2015)."

Therefore, these data are not really comparable and you need to discuss your data accordingly. We agree that it is not a beta diversity analysis; we are going to consider thresholds, and modulate the sentence by adding:

A similar observation was made in eastern China, with different spatial distributions of rare and dominant soil fungal OTUs (the authors did not use Hill numbers but relative abundance and a threshold to group rare and dominant OTUs), …

Line 462-468: Accordingly, to the previous comment, "national distribution of dominant OTUs" is not correct, but rather "dominant OTUs per sample". Do these references define dominant OTUs in the same way you do it? Is it a valid comparison? Adapt accordingly in the entire discussion.
We agree, please see previous comments.

Line 470: What do you mean with trophic condition? Please, specify.
Please see our text in parentheses: …  '(organic C and total Fe contents).'

Line 476: Instead of "positive or negative effect" use "positive or negative linear relationship".
We agree, we are going to change with "positive or negative linear relationship".

Line 479: What did they find at the European scale with this large gradient? This argument is confusing.
We agree, we are going to remove it to clarify.

Line 484: Richness is not biomass. Higher richness does not mean that there are more fungi, but more different types of fungi. So the argument, that fine-textured soil offer less habitat and therefore lower richness is expected does not make sense.
We agree, we are going to replace 'favorable' by 'diverse' to clarify.
In France, we may think that fine-textured soils offer less diverse habitats for fungi, as previously reported (Witzgall et al., 2021; Tecon and Or, 2017).

Line 498: I do not think that "whatever the hill number" is a correct expression in English, please adapt.
We are going to change the sentence to clarify it.
… of the distribution of fungal diversity across France for all Hill numbers based on three different orders of diversity (Terrat et al., 2017).

Line 505: See comment from line 458-461.
We agree, please see previous comment.

Line 511: What do you consider "more or less important human activities"? Important for what / whom? Please specify.
We are going to change the sentence.
In France, each land use corresponds to a particular intensity of soil disturbance resulting from the intensification of agricultural activities.

Line 512: Why do you think that vineyards and orchards are more disturbed than crop land?
We are going to add a sentence to clarify.
The crop systems – vineyards in particular – use pesticides and soil tillage that have deleterious effects on fungal diversity (Karimi et al., 2021; Christel et al., 2021).

Karimi, Battle, et al. "Ecotoxicity of copper input and accumulation for soil biodiversity in vineyards." Environmental Chemistry Letters 19 (2021): 2013-2030.
Christel, Amélie, Pierre-Alain Maron, and Lionel Ranjard. "Impact of farming systems on soil ecological quality: a meta-analysis." Environmental Chemistry Letters 19.6 (2021): 4603-4625.

Line 527-530: What surprised you about this finding? Please explain.
We are going to add a sentence to clarify.
This observation suggests that increased fungal diversity in intensive soil management crop systems is concurrent with decreased fungal evenness.

Line 557: This is not an "interaction network" but a co-occurrence network. I think we need to use these terms more carefully in the field of microbial ecology.
We agree, we are going to change 'interaction network' by 'co-occurrence network' throughout the manuscript to clarify.

Line 565: Are vineyards regularly and intensively tilled in France? I did not find this information in the provided reference.
We agree, we are going to add the reference Karimi et al. (2020).
Karimi, Battle, et al. "A meta-analysis of the ecotoxicological impact of viticultural practices on soil biodiversity." Environmental Chemistry Letters 18 (2020): 1947-1966.

Line 573-575: The alpha diversity data suggest a higher diversity in crop lands, wouldn't that also be a reason to call crop land a "favorable habitat"?
We do not think so because it was very recently showed that systems more disturbed than forests (i.e., grasslands and crops) harbored higher richness but also a higher proportion of fungal plant pathogens (Labouyrie et al., 2023).
Labouyrie, Maëva, et al. "Patterns in soil microbial diversity across Europe." Nature Communications 14.1 (2023): 3311.

Conclusions: Only few and very broad and general conclusions were included. It would be a pity to omit more specific conclusions on fungal communities, especially as they were assessed in such a huge sampling system.

We completed the Conclusion by adding the following text:
At the scale of France, the soil fungal diversity is driven by soil characteristics, land management and climatic conditions. Soil pH was the most important soil property explaining rare and abundant fungal diversity. The lowest fungal richness was found in less disturbed environments (forests) and highly disturbed environments (vineyards & orchards) compared to grasslands and croplands. Highly disturbed environments (crops, vineyards & orchards) harbored the lowest fungal network complexity compared to forest soils, which harbored the most connected networks.